# CONCEPT ALIGNMENT AS A PREREQUISITE FOR VALUE ALIGNMENT

## ABSTRACT

Value alignment is essential for building AI systems that can safely and reliably interact with people. However, what a person values—and is even capable of valuing—depends on the concepts that they are currently using to understand and evaluate what happens in the world. The dependence of values on concepts means that *concept alignment* is a prerequisite for value alignment—agents need to align their representation of a situation with that of humans in order to successfully align their values. Here, we formally analyze the concept alignment problem in the inverse reinforcement learning setting, show how neglecting concept alignment can lead to systematic value mis-alignment, and describe an approach that helps minimize such failure modes by jointly reasoning about a person's concepts and values. Additionally, we report experimental results with human participants showing that humans reason about the concepts used by an agent when acting intentionally, in line with our joint reasoning model.

## 1 INTRODUCTION

People's thoughts and actions are fundamentally shaped by the concepts they use to represent the world and formulate their goals. Imagine watching someone waiting to cross a busy intersection. Making sense of their behavior requires understanding their representation of things like "the crosswalk," "the road," "the bike lane," and "the right of way." For instance, it is important to take into account whether someone understands or is aware of the part of the street designated the "bike lane" while they wait since otherwise their intentions could be misinterpreted (e.g., a naïve observer might think someone standing in the bike lane is *trying* to get hit by a bicycle). Yet, current approaches to inferring human goals, rewards, and values (e.g., standard inverse reinforcement learning (Abbeel & Ng, 2004) and value alignment (Hadfield-Menell et al., 2016) largely neglect the possibility that a machine observer and a human actor can have misaligned concepts. Our goal in this work is to formally state the problem of *concept alignment*, begin to explore algorithmic solutions, and compare these solutions to human judgments.

To formalize concept alignment, we draw on the recently proposed framework of *value-guided construal* (Ho et al., 2022), which provides a computational account of how humans form simplified representations of problems in order to solve them. A *construal* is a particular interpretation of a problem in terms of a set of concepts and related causal affordances. Different construals encode different conceptual understandings of the world: for example, if one understands the concept of the bike lane and includes it in their current construal, they are aware that bicycles are often in the bike lane, cars generally avoid the bike lane, you might get hit if you stand in the bike lane, etc. People often prefer simpler construals since they are less cognitively effortful (Ho et al., 2023), but this can affect the quality of one's actions—e.g., if you fail to distinguish the bike lane from the sidewalk, you might stand in a place where a bicycle will hit you! As we discuss later, our approach is to incorporate construals into a forward model of planning, which allows us to articulate the problem of conceptual misalignment as a form of misspecified *inverse* planning (Baker et al., 2009).

We propose a theoretical framework for formally introducing concepts to inverse reinforcement learning and show that conceptual misalignment (i.e., failing to consider construals) can lead to severe value misalignment (i.e., reward mis-specification; large performance gap). We validate these theoretical results with a case study using a simple gridworld environment where we find that IRL agents that jointly model construals and reward outperform those that only model reward. Finally,

we conduct a study with human participants and find that people do model construals, and that their inferences about rewards are a much closer match to the agent that jointly models construals and rewards. Our theoretical and empirical results suggest that the current paradigm of just trying to directly infer human reward functions or preferences from demonstrations is insufficient for value-aligning real AI systems that need to interact with real people; it is crucial to also model and align on the concepts people use to reason about the task in order to understand their true values and intentions.

## 2 RELATED WORK

Work on inferring human preferences and values is often done in the framework of inverse-reinforcement learning (IRL) (Abbeel & Ng, 2004; Hadfield-Menell et al., 2016; Ho et al., 2016) and inverse planning (Baker et al., 2009). In the standard IRL setting, an agent is tasked with estimating or inferring the reward function that an expert is optimizing. An important benefit of IRL over other methods for learning from expert human behavior, such as behavioral cloning (Munro et al., 2011), is that it facilitates *generalization* to new scenarios outside of the data given. For instance, by inferring that a human has a dispreference for eating spinach after observing behavior at home, an agent could anticipate behavior in new scenarios in which spinach appears, such as in a restaurant. Over the past two decades, methods for IRL have been extended in various ways and even used as models for social cognition in cognitive science (Ho & Griffiths, 2022).

However, a key property of virtually all existing IRL methods is that they assume behavior emerges from a planning process that produces optimal or noisy-optimal policies (Abbeel & Ng, 2004; Loftin et al., 2016; Ziebart et al., 2008). This assumption is problematic because it is false (Simon, 1955; Tversky & Kahneman, 1974). An alternative perspective that has been developed over the past few years is that people are *resource-rational*—that is, they think and act rationally, but are subject to cognitive limitations on time, memory, or attention (Lieder & Griffiths, 2020). A major research challenge for IRL, value alignment, and cognitive science is incorporating these ideas into estimating human preferences and values (Ho & Griffiths, 2022; Evans et al., 2016; Zhi-Xuan et al., 2020; Kwon et al., 2020; Alanqary et al., 2021; Chan et al., 2021; Laidlaw & Dragan, 2022).

The work here builds on recent approaches to modeling resource-rational human planning in the *value-guided construal* framework, which provides an account of how humans rationally simplify problems and apply simplified concepts in order to plan (Ho et al., 2022; 2023). The key idea of value-guided construals is that people do not necessarily use all concepts available when representing a problem in order to make efficient use of limited attention (e.g., ignoring certain details of obstacles when navigating through a GridWorld). Applied to the IRL setting, this involves inverting the value-guided construal model of human decision-making and using it instead of the classical noisy-rational model. Our goal here is to provide an initial demonstration of the utility of incorporating concept simplification strategies into value alignment and IRL.

## 3 INTRODUCING CONSTRUALS INTO INVERSE REINFORCEMENT LEARNING

We begin by reviewing the basic formalism for sequential decision-making before turning to construals and the inverse planning problem.

### 3.1 BACKGROUND

We represent sequential decision-making tasks as Markov decision-processes (MDPs) $M = \langle \mathcal{S}, \mathcal{A}, P_0, T, R, \gamma \rangle$, where $\mathcal{S}$ is a state space; $\mathcal{A}$ is an action space; $P_0 : \mathcal{S} \to [0, 1]$ is an initial state distribution; $T : \mathcal{S} \times \mathcal{A} \times \mathcal{S} \to [0, 1]$ is a transition function; $R : \mathcal{S} \times \mathcal{A} \to \mathbb{R}$ is a real-valued reward function; and $\gamma \in [0, 1)$ is a discount rate. A (stochastic) policy is a conditional probability distribution that maps states to distributions over actions, $\pi : \mathcal{S} \to \Delta(\mathcal{A})$. We denote the Markov chain resulting from following policy $\pi$ on an MDP with dynamics $T$ as $T^\pi(s' \mid s) = \sum_a \pi(a \mid s) T(s' \mid s, a)$.

We consider standard (unregularized) and entropy-regularized solutions to MDPs. In the unregularized setting, the value function associated with a policy $\pi$ on an MDP with dynamics $T$ and reward function $R$ maps each state to the expected cumulative, discounted reward that results from following

$\pi$: $V_{(R,T)}^{\pi}(s) = \sum_a \pi(a \mid s)[R(s,a) + \gamma \sum_{s'} T(s' \mid s,a)V_{(R,T)}^{\pi}(s')]$. The state occupancy function (also known as the successor representation) associated with a policy $\pi$ on an MDP with dynamics $T$ is the expected discounted visitations to a state $s^+$ starting from a state $s$, $\rho_T^{\pi}(s; s^+) = \mathbf{1}[s^+ = s] + \gamma \sum_{s'} T^{\pi}(s' \mid s)\rho_T^{\pi}(s'; s^+)$. The optimal value function for an MDP $M$ maximizes value at each state, $V_{(R,T)}^{*}(s) = \max_a\{R(s,a) + \gamma \sum_{s'} T(s' \mid s,a)V_{(R,T)}^{*}(s')\}$.

In the entropy-regularized setting, the value of a policy $\pi$ on MDP $M$ is modified to include an entropy term, which penalizes action distributions that are more deterministic: $H(\pi(\cdot \mid s)) = -\sum_a \pi(a \mid s)\ln\{\pi(a \mid s)\}$. When this penalty is parameterized by a weight $\beta$, we denote the optimal entropy-weighted value function as $V_{(R,T)}^{\beta}(s) = \max_{\pi}\{\sum_a \pi(a)[R(s,a) + \sum_{s'} T(s' \mid s,a)V_{(R,T)}^{\beta}(s')] + \beta H(\pi)\}$.

## 3.2 INVERSE REINFORCEMENT LEARNING (IRL)

The standard IRL problem formulation involves an *observer* attempting to estimate the reward function of an expert *demonstrator* based on observed behavior. This can be formalized as Bayesian inference, where given a trajectory of expert acting in the task, $\zeta = \{\langle s_0, a_0 \rangle, \langle s_1, a_1 \rangle, ..., \langle s_T, a_T \rangle\}$, the observer infers the demonstrator's reward function, $R$:

$$P(R \mid \zeta) = \frac{P(\zeta \mid R)P(R)}{P(\zeta)}. \tag{1}$$

To calculate the likelihood of a trajectory $\zeta$ given a reward function $R$, it is typically assumed that the observer has knowledge of the dynamics of the demonstrator's task, $T$. Then, the likelihood is the probability of the trajectory being generated by the optimal policy under a candidate $R$:

$$P(\zeta \mid R) = \prod_{\langle s_t, a_t \rangle \in \zeta} \pi_{(R,T)}^{\beta}(a_t \mid s_t). \tag{2}$$

## 3.3 INVERSE CONSTRUAL

The inverse construal problem considers the possibility that although a resource-limited demonstrator is acting in a task with a particular dynamics $T$, they may not be *planning their actions* with respect to the fully-detailed dynamics. Rather, the demonstrator's behavior results from planning with respect to a *construed task dynamics*, $\tilde{T}$, which reflects their understanding (or lack thereof) of notches as traversable.

Thus, an observer that takes into account the resource limitations faced by human planners should instead be aiming to solve an inference problem that incorporates the possibility of alternative task construals. Formally, this is the problem:

$$P(R, \tilde{T} \mid \zeta) = \frac{P(\zeta \mid R, \tilde{T})P(R, \tilde{T})}{P(\zeta)}, \tag{3}$$

where the prior $P(R, \tilde{T})$ is uniform and the likelihood is given by

$$P(\zeta \mid R, \tilde{T}) = \prod_{\langle s_t, a_t \rangle \in \zeta} \pi_{(R,\tilde{T})}^{\beta}(a_t \mid s_t). \tag{4}$$

## 3.4 CONSEQUENCES OF NOT CONSIDERING CONSTRUALS

How bad can the estimate of $R$ be when assuming the true dynamics $T$ versus attempting to estimate the demonstrator's construal $\tilde{T}$? If we use a maximum causal entropy formulation of IRL to get an estimated policy $\hat{\pi}^{\text{InvRL}}$ and compare this to the estimated policy assuming the demonstrator is using a construal, $\hat{\pi}^{\text{InvCon}}$, then the learner's performance gap on the true task is (Viano et al., 2021):

$$|v_{(R,T)}^{\hat{\pi}^{\text{InvCon}}} - v_{(R,T)}^{\hat{\pi}^{\text{InvRL}}}| \leq \frac{\gamma \cdot |R|^{\max}}{(1-\gamma)^2} \cdot \max_{s,a} ||T(\cdot \mid s,a) - \tilde{T}(\cdot \mid s,a)||_1$$

where $|R|^{\max} = \max_{s,a} |R(s,a)|$. This is a tight bound, and thus the risk associated with not modeling construals (i.e., the potential size of the gap) grows rapidly when either the discounting, the

maximum reward, or the task mismatch increases. In other words, if the observer has an inaccurate estimate of the transition function the actor uses to plan, they may drastically mis-estimate the reward function that motivated behavior. This provides a formal expression of our introductory example, in which failing to consider that a person does not know about or is unaware of a bike lane might lead one to interpret standing in the bike lane as indicating a *desire* to be hit by a bicycle.

## 4    A SIMPLE EXAMPLE OF CONCEPT MISALIGNMENT

Our theoretical results show that concept misalignment creates risk of value misalignment (i.e, a large performance gap *can* exist). We now aim to show that the performance gap indeed exists in practice.

First, we flesh out our bike lane example into a city navigation case study. Suppose Alice is trying to navigate a city to get a cup of coffee. There is a mom-and-pop bakery where she could get her favorite pastry and a delicious coffee, and a fast food franchise where she will have to wait in line and overpay but will still get a decent coffee. Given the choice, she would strongly prefer to go to the bakery. There are areas she can walk through (i.e., streets) and areas she cannot go through (i.e., locked buildings). However, there are also some unlocked buildings that she could cut through if only she knew that they are unlocked. If she perceives both the bakery and the fast food place as being accessible, she will always choose to go to the bakery (regardless of distance). However, if only the latter seems accessible, she will go get her coffee there. We visualize this setup in Figure 1. The left column corresponds to this hypothetical case where Alice prefers the bakery, and the right column corresponds to an alternate hypothetical case where Alice instead prefers the fast food place. Now consider the case where the bakery is inside of a closed courtyard and the only way to reach it is to go through an unlocked building, but the fast food place is outside of the courtyard. If Alice is unaware that there are unlocked buildings that give access to the courtyard, she may end up going to the fast food place. An observer who does not take into account that Alice does not realize there are unlocked buildings to cut through would incorrectly infer that she prefers the fast food place (see Figure 1 bottom-left).

To investigate the impact of modeling (or not modeling) a construal on value alignment between a human demonstrator and a machine IRL agent in scenarios such as this city navigation example, we use *blocks and notches* maze tasks similar to those developed by Ho et al. (2023) to study rigidity in people's construals (Figure 1). Each blocks-and-notches maze consists of a start state depicted as a blue circle (e.g., where Alice starts off), a high-value goal (e.g., the bakery) depicted as a pink or yellow square, a low-value goal (e.g., the fast food place) depicted as a yellow or pink square, and blue $3 \times 3$ blocks (e.g., buildings). The dark blue squares prevent movement (e.g., locked buildings), but the light blue notches (e.g., unlocked buildings) permit movement through the blocks. This environment allows us to simulate scenarios analogous to the city navigation ones described above.

### 4.1    NOTCHES

In our simulations, notches (represented by light blue squares within the $3 \times 3$ blue blocks) are shortcuts through the grid. The idea is that while everyone is shown the same view when tasked with navigating the grid, only some demonstrators notice and learn how to use the notches; others ignore the light blue vs dark blue distinction and treat the entire $3 \times 3$ blue block as an obstacle, which they navigate around. In other words, people with different construals of the same ground truth grid learn different paths (Ho et al., 2023).

A standard IRL agent trying to infer a demonstrator's rewards in this task is misaligned at the concept level because it assumes an optimal policy (and therefore has no notion that a human might not understand notches or how to use them). Humans, as we have discussed before, often act in ways that are not conventionally considered optimal or even rational. The IRL agent, without an understanding of the different construals people are using to understand the grid, draws incorrect conclusions about people's values (rewards).

Of the four scenarios (Figure 1) used in our experiments, the two on the bottom depict routes taken by simulated demonstrators who did not realize they could walk through notches. The near (pink) goal is unreachable without using notches; think of it as the bakery enclosed on all sides by buildings

($3 \times 3$ dark blue blocks) some of which are unlocked (light blue notches). The grids on the top show the trajectory of a simulated demonstrator who has learned that a notch is a shortcut, and has used the notch to form a more efficient path to their preferred goal. On the bottom are the trajectories of a simulated demonstrator who only knows the blue $3 \times 3$ blocks are obstacles, without paying attention to the fact that some sub-blocks (notches) are not obstacles at all. Looking at these trajectories on the lower half, the IRL agent which does not have any notion of construals and assumes an optimal policy would naturally assume the human demonstrator has a value-related reason for avoiding the pink goal, and would thus assume that the yellow goal has a higher reward. Thus we see value misalignment emerge as a consequence of concept misalignment between the human demonstrator and the IRL agent.

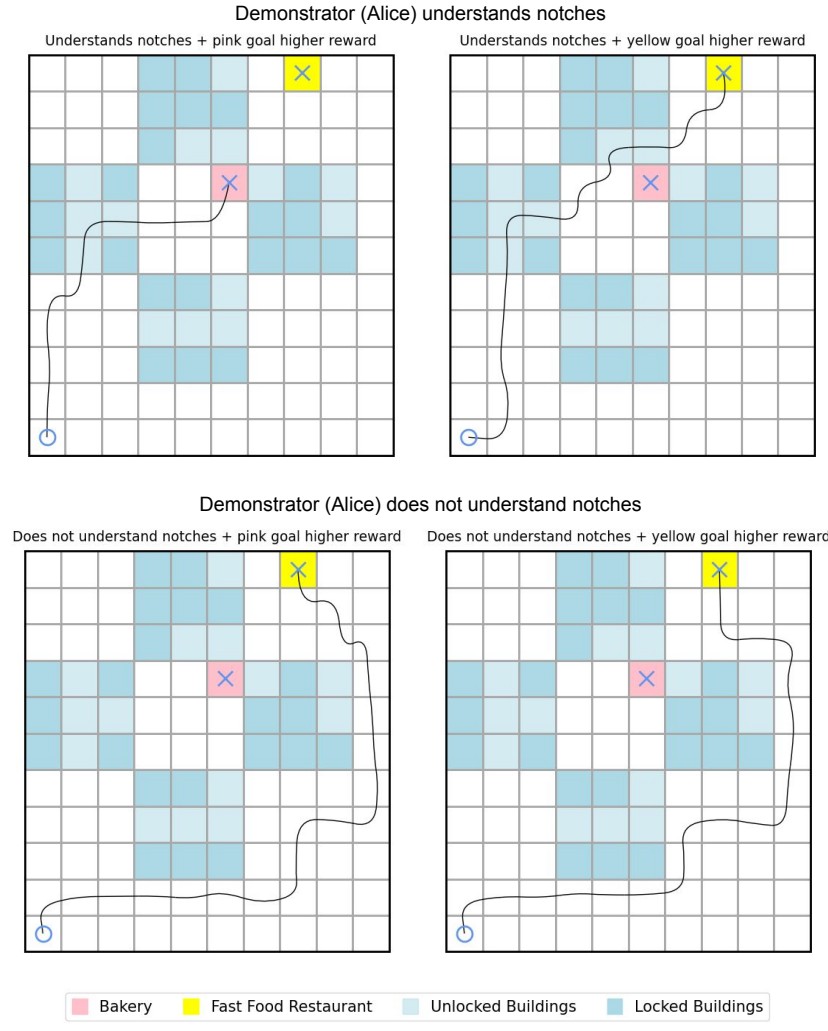

Figure 1: Four trajectories produced by different combinations of rewards and construals. The two trajectories on the lower half with the construal "Does not understand notches" look similar, because the preferred (pink) goal is impossible to reach when not construing notches.

## 4.2 VALUE MISALIGNMENT

In our reinforcement learning framework, we use rewards as a proxy for values. To demonstrate how concept misalignment can lead to value misalignment between humans and machines, we employ an inverse reinforcement learning agent to infer the human's values (reward function). Without

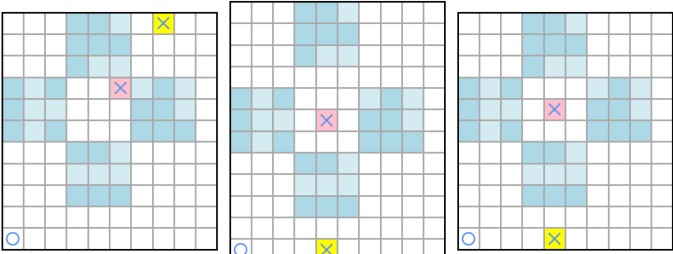

Figure 2: The three GridWorlds used in our experiments.

knowledge of the construals (different understanding of notches), the agent might misattribute the path to a higher reward value for the chosen goal, not realizing the other goal may in fact have a higher reward, but may be impossible to reach without using/paying attention to notches.

As a measure of how alignment at the construal level can improve value alignment, we compare the posterior probability $P(R, \tilde{T} \mid \varsigma)$ when jointly modeling the reward and the construal, to $P(R \mid \varsigma)$, the standard IRL posterior which assumes that the trajectory is coming from a policy optimal with respect to the true transition function.

We run inference to calculate these probabilities for three GridWorlds shown in Figure 2 using both the reward-only model and the joint reward and construal model. The full twelve trajectories used for inference (four scenarios over each of the three GridWorlds) are shown in Figure 4.

### 4.3 MODEL RESULTS

We find that the joint reward and construal model performs on par with the reward-only model in the settings where the demonstrator "understands notches", but significantly outperforms the reward-only model in inferring values in the difficult "Does not understand notches" scenario where the preferred (pink) goal is inaccessible without using notches. In this scenario, the joint model correctly infers that the pink goal has a higher reward even though the demonstrator visits it in only one out of the three demonstrations for that scenario. The reward-only model fails completely, inferring confidently and incorrectly that the yellow goal has a higher reward due to the higher number of visits. See Figure 4 for a full comparison.

## 5 HUMAN EXPERIMENTS

We have now shown both theoretically and in simulation that a performance gap due to concept misalignment is not only possible but also plausible. But if vanilla inverse reinforcement learning is insufficient for inferring people's intent and preference in the real world, then how do people manage to do these things in practice? We now show that humans a) are highly adept at reasoning about construals, and b) use their knowledge of construals when making inferences about others' paths.

In this behavioral experiment, we gave 100 human participants the same four trajectories given to the two IRL agents (Figure 1) and asked them to make the same inferences. Each participant was shown a live replay of each trajectory, and then asked to infer (Figure 3) whether the person who took this route realized they could walk through notches, and if they preferred the pink goal to the yellow goal. Participants were asked to respond true or false to each question, and these responses were then mapped to scores of 1 or -1 when computing the results. These two questions map naturally to the posterior of the IRL algorithm's construal and reward inferences about each goal.

A full walkthrough of instructions, visuals, and questions shown to the human participants is included in the supplementary materials [1]. We also scale the IRL posterior inferences to this -1 to 1 scale for direct comparison with the human judgments.

---

[1]Code and data: `https://osf.io/hd9vc/?view_only=967b0c2f981d4a87bf4d21ff818f1322`

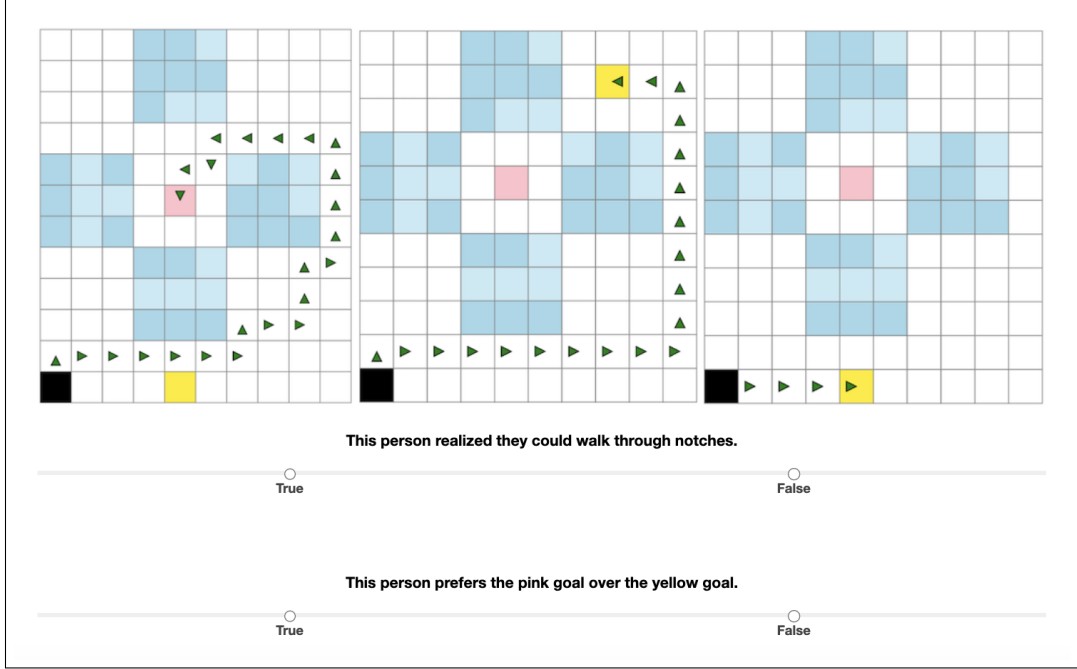

Figure 3: One frame of the data collection process where we collected human judgements on the IRL task given the trajectories in Figure 1. In this example, the person prefers the pink goal but does not realize they can walk through notches. In the leftmost grid, they can access their preferred pink goal without using notches. But in the other two grids, they cannot access their preferred pink goal and opt for the yellow goal instead. An IRL algorithm that does not consider construals would assume that the person is intentionally choosing the yellow goal over the pink most of the time, and miscalculate the yellow goal having a higher reward, when in fact it is the other way around.

Table 1: Proportion of human participants who correctly inferred rewards for each scenario. p-values are from a one-sided binomal test corresponding to the null hypothesis that the human reward inferences are explained by random chance.

| Scenario | Proportion correct | p-value |
|---|---|---|
| Understands notches + pink goal higher reward | 0.99 | 7.967e-29 |
| Doesn't understand notches + pink goal higher reward | 0.70 | 3.925e-05 |
| Understands notches + yellow goal higher reward | 0.98 | 3.984e-27 |
| Doesn't understand notches + yellow goal higher reward | 0.98 | 3.984e-27 |

## 5.1 RESULTS

There are three components to our results: human data, IRL inference when jointly modeling rewards and construals, and IRL inference when modeling only reward. These results are shown side-by-side in Figure 4. The posteriors of the IRL inference are scaled to match the -1 to 1 scale of the human data. Error bars for human data are one standard error from the mean over all 100 participants, for each question of each trajectory.

We show that humans completing the same inference task as the IRL agents successfully use construals to make more accurate reward inferences (see Table 1), matching the behavior of the joint reward and construal model (see Figure 4). We also calculate correlations between human reward inferences and model reward inferences, which demonstrate that the joint reward and construal model is highly correlated with human reward inferences in the same scenarios (see Table 2).

Table 2: Correlations between model and human inferences of reward

|  | Pearson correlation coefficient | p-value |
|---|---|---|
| Reward-only IRL | 0.757 | 0.242 |
| Jointly-modeled IRL | **0.970** | **0.029** |

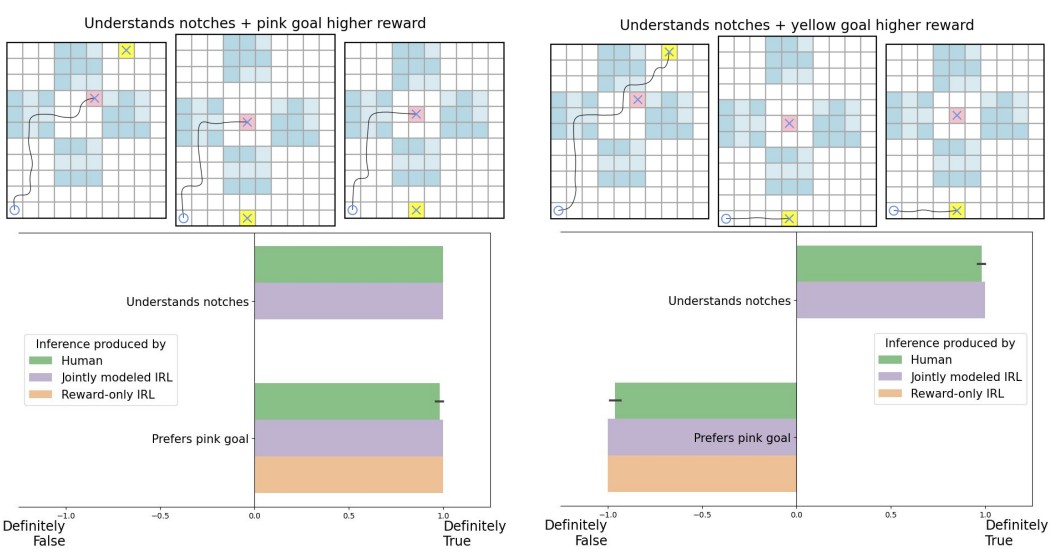

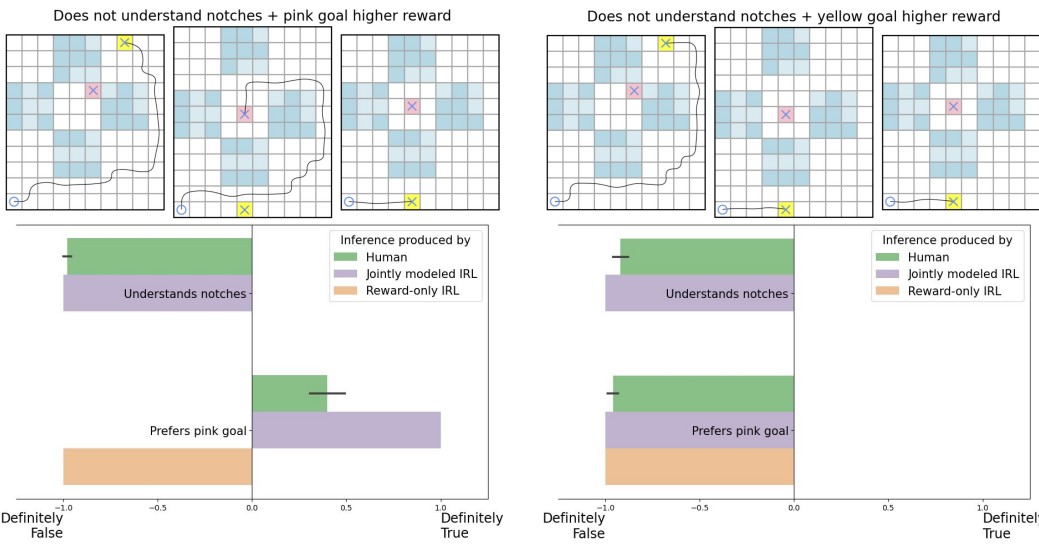

Figure 4: Inferences produced by humans and the two models. In the most difficult "Does not understand notches" scenario in the lower left corner, jointly modeling construals and rewards allows the IRL algorithm to successfully infer that the pink goal has higher reward, despite not being the most frequently accessed goal. Human subjects also make this inference. The reward-only IRL agent answers incorrectly and confidently that the yellow goal has higher reward.

# 6    DISCUSSION

In this work, we formulate the problem of concept alignment within the framework of value-guided construals. When people are faced with a task, they often do not represent it in full detail and instead engage in simplification strategies to make more efficient use of limited cognitive resources (Ho et al., 2022). As a result, people may use simplified concepts that lead to different behaviors than if they had represented the task in complete detail. Our main goal here has been to formalize the inverse problem of estimating what simplified concepts people are using and show how such an approach is needed for successful value alignment in a simple setting.

In the most difficult scenario of the "Does not understand notches" construal, the reward-only IRL agent confidently makes a very incorrect inference (see Figure 4, because it does not model the construal.

Modeling construals and allowing for alignment at a conceptual level enables the IRL algorithm to correctly infer human rewards and values instead of confidently making an incorrect inference. Modeling construals also brings the IRL behavior closer to the human participants' behavior, because it creates a shared conceptual framework which enables more accurate reasoning about another person's rewards and values.

## 6.1    SOCIETAL IMPLICATIONS OF CONCEPT MISALIGNMENT

Our results carry important implications for almost all settings where AI systems are expected to interact with people and align with their values or preferences, often in high-stakes, complex scenarios. For example, scaling up concept alignment to a healthcare diagnostic algorithm would not require agent-based IRL at all. Rather the concepts would be the medical imaging artifacts supporting diagnostic decisions, labelled by the words human physicians use to describe those artifacts. The goal would be to align those concepts to a diagnostic algorithm which analyzes the medical image and suggests a course of action, like surgery. This type of concept alignment would be a necessary prerequisite to meaningfully discuss surgery based on what the patient values most (mobility, lack of pain, effect on other organs, long-term psychological impact of the procedure). In many such contexts of AI-human interaction, concept-level alignment lays the groundwork for value alignment.

## 6.2    LIMITATIONS AND FUTURE WORK

While our theoretical results apply to many different settings with different types of construals, our experiments focused on a case study where we could control the features that form the basis for construals. In real-world settings, there are countless more features and a simple construal model would clearly be insufficient. However, our goal was not to suggest that this specific simplified construal model is the solution to value alignment, but rather to demonstrate that concept misalignment is, in fact, a problem that AI researchers need to focus on to make progress on value alignment. In future work, we intend to scale this approach in a wider variety of settings with human experts, and involving a wider variety of inference algorithms that can scale to larger reward and construal spaces (Ho & Ermon, 2016; Herman et al., 2016; Chakraborti et al., 2019; Gopalakrishnan et al., 2021; McCallum, 1996; Cao et al., 2021). As the algorithms scale, so too should the empirical studies of human behavior to measure their alignment at the concept- and value-levels. More broadly, we hope that demonstrating the critical importance of concept alignment to the larger goal of value alignment will open the door to future work characterizing concept and value alignment in real-world settings.

## 6.3    CONCLUSION

Human decision-making is complex, context-driven, and resource-dependent and we have to keep that in mind when we try to teach AI systems to infer human values. We have laid out a framework introducing the notion of concepts into inverse reinforcement learning and showed both theoretically and empirically that without concept alignment it may often be impossible to achieve value alignment. We also showed in a behavioral study that people reason about each other's concepts when making inferences about each other's goals and values. We hope that these results encourage other researchers to work on concept alignment as a crucial component in value alignment, effective human-AI interaction, and the development of safe autonomous agents.

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

# A    APPENDIX A: HUMAN EXPERIMENT WALKTHROUGH

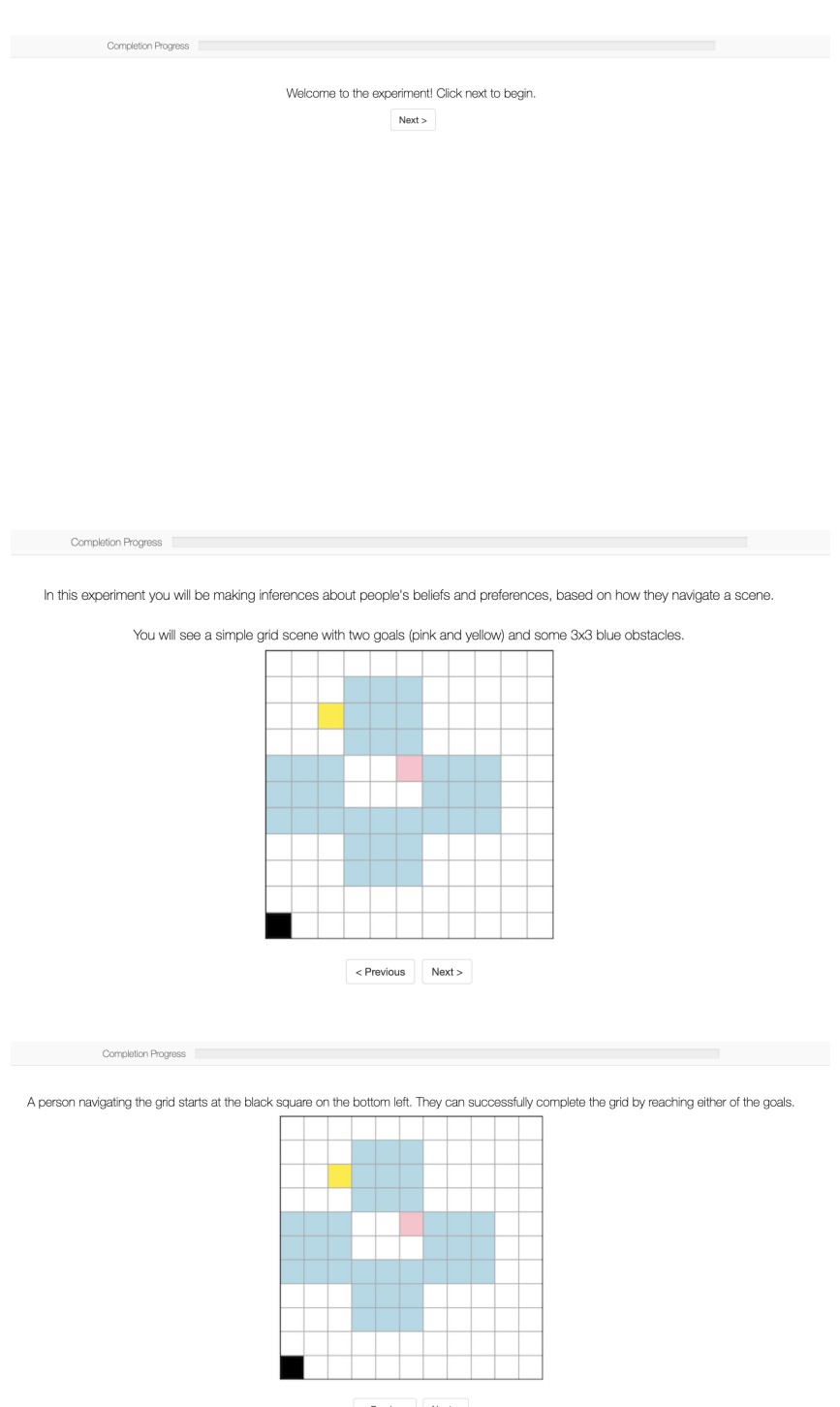

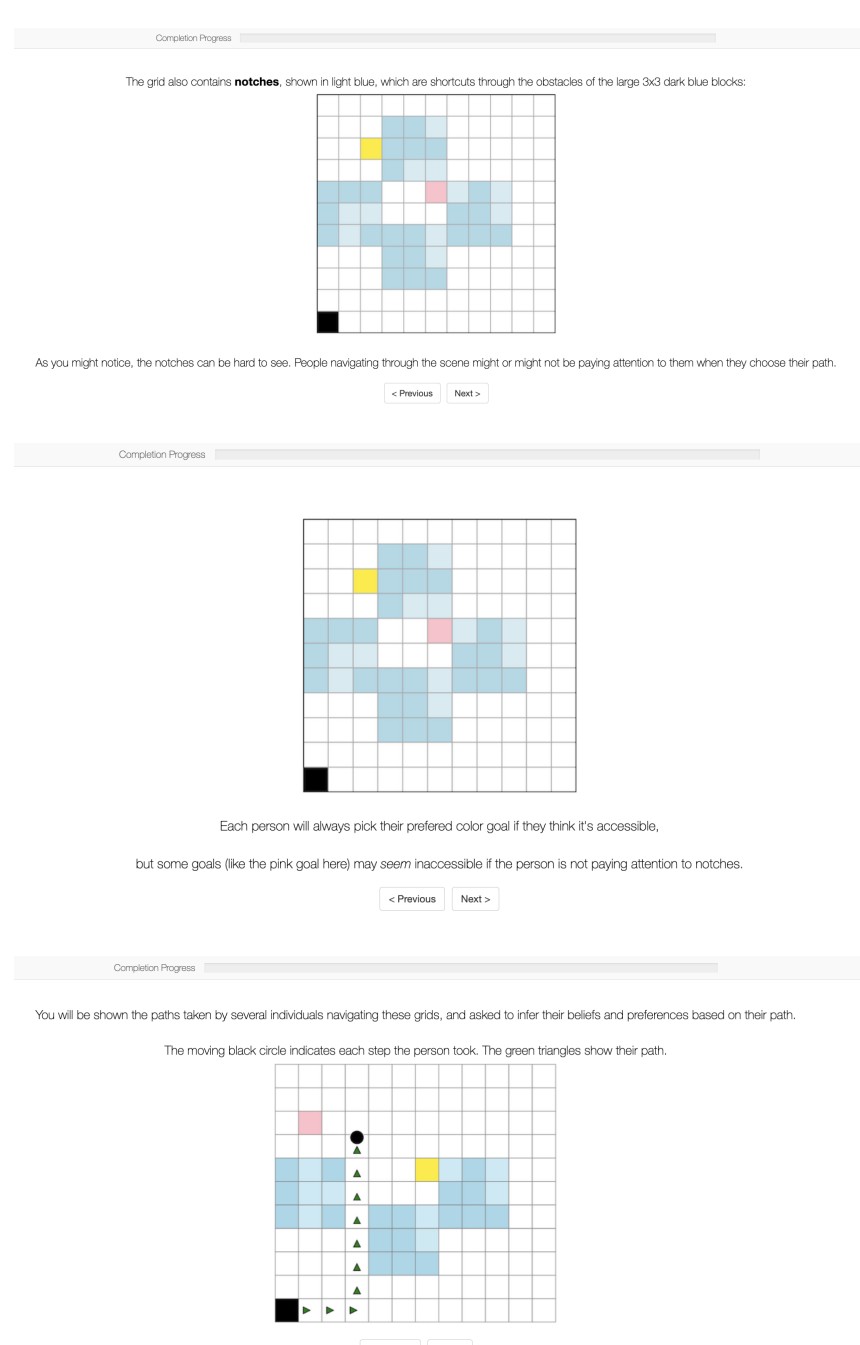

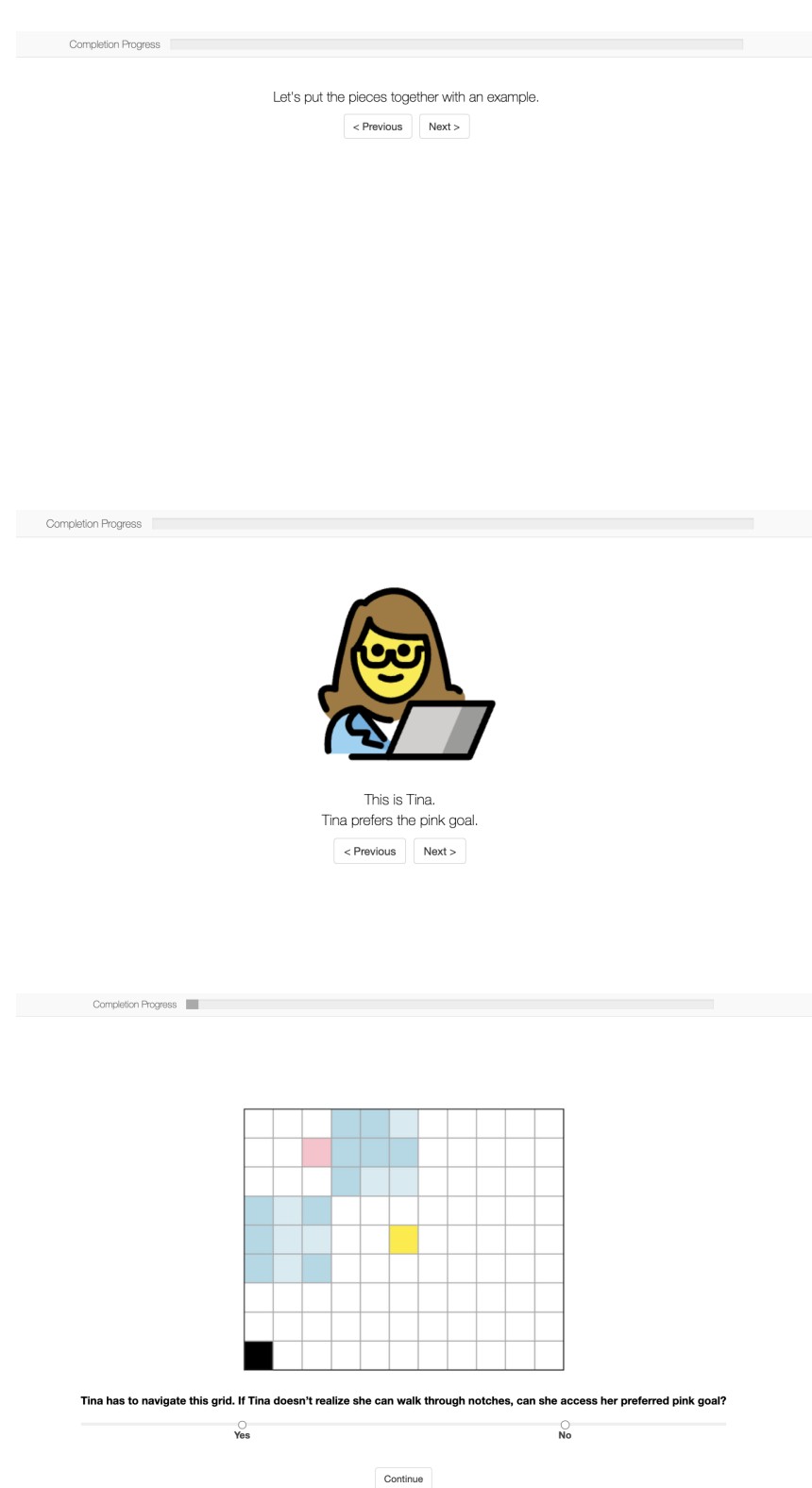

Completion Progress

Let's put the pieces together with an example.

< Previous    Next >

Completion Progress

This is Tina.
Tina prefers the pink goal.

< Previous    Next >

Completion Progress

**Tina has to navigate this grid. If Tina doesn't realize she can walk through notches, can she access her preferred pink goal?**

○ Yes          ○ No

Continue

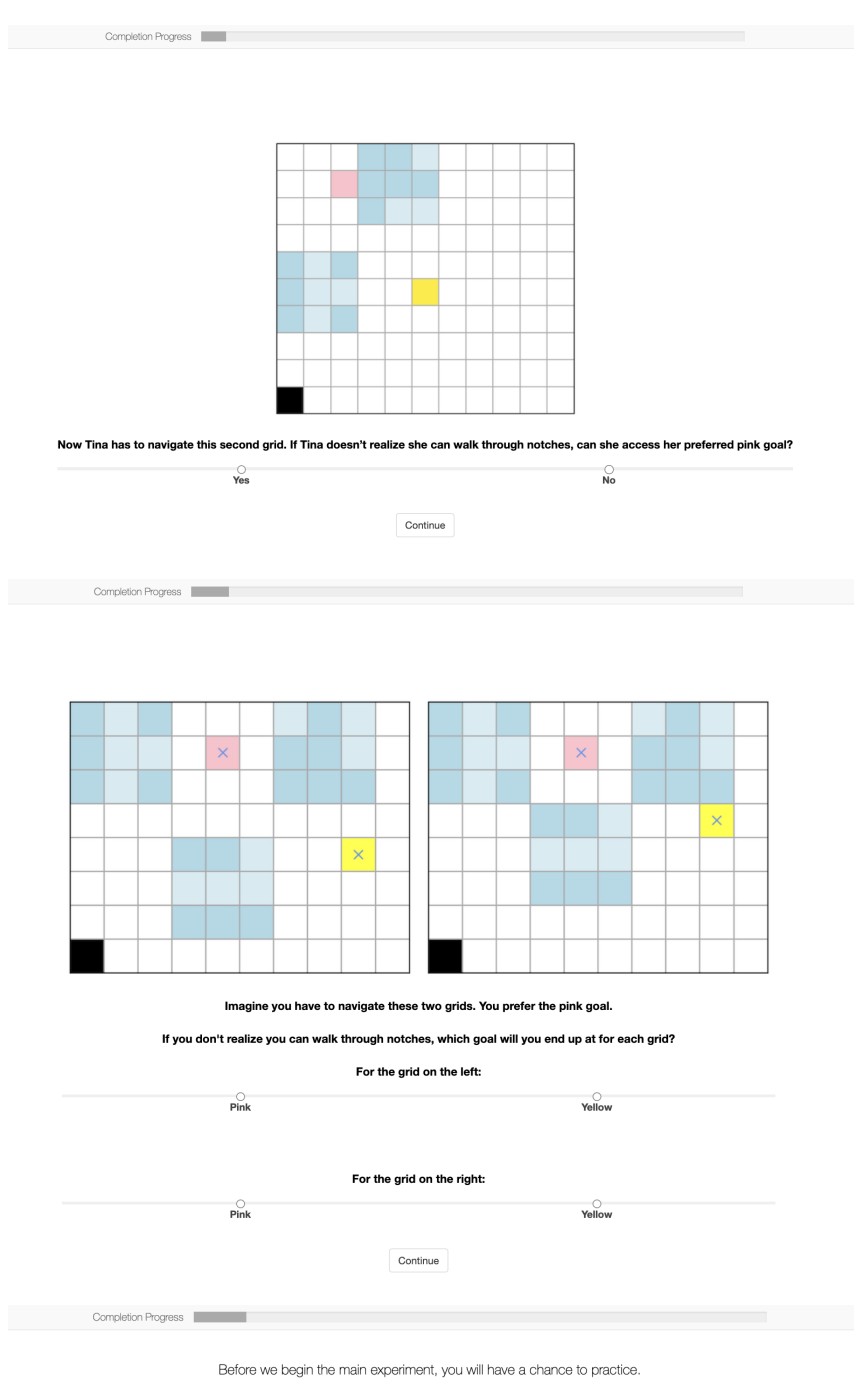

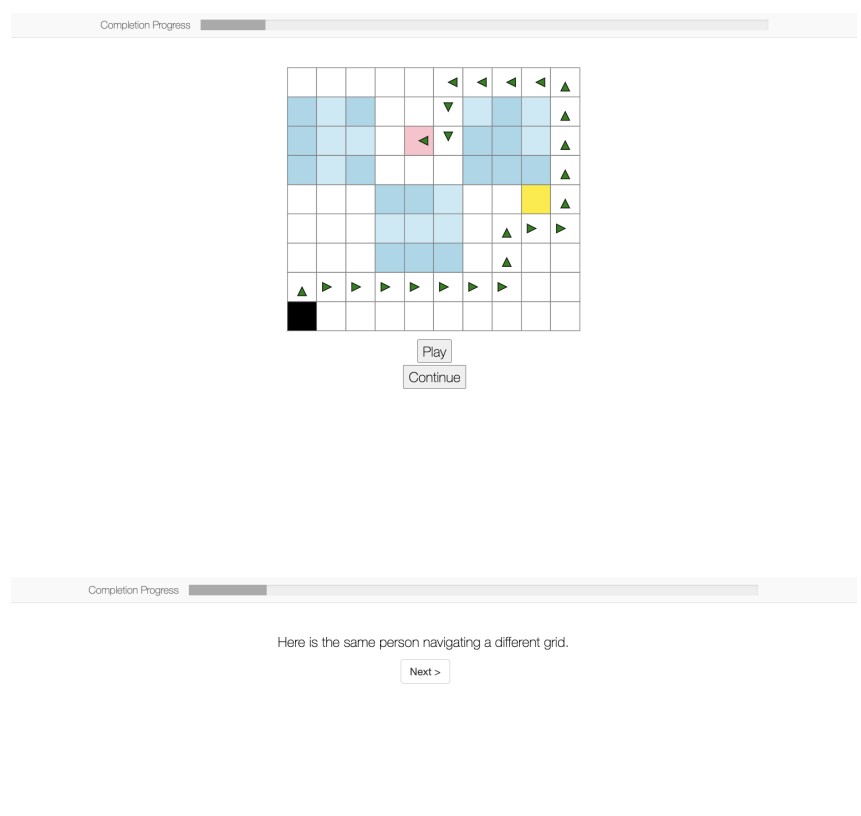

Here is the same person navigating a different grid.

Next >

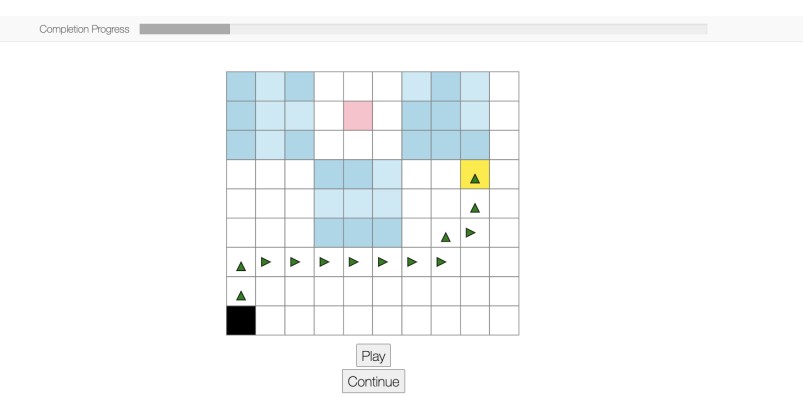

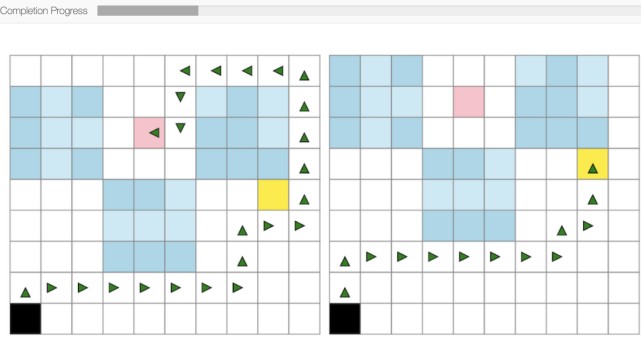

Now please answer these questions about this person's beliefs and preferences.

If you need to, you can scroll up to revisit what the completed trajectory looked like.

Next >

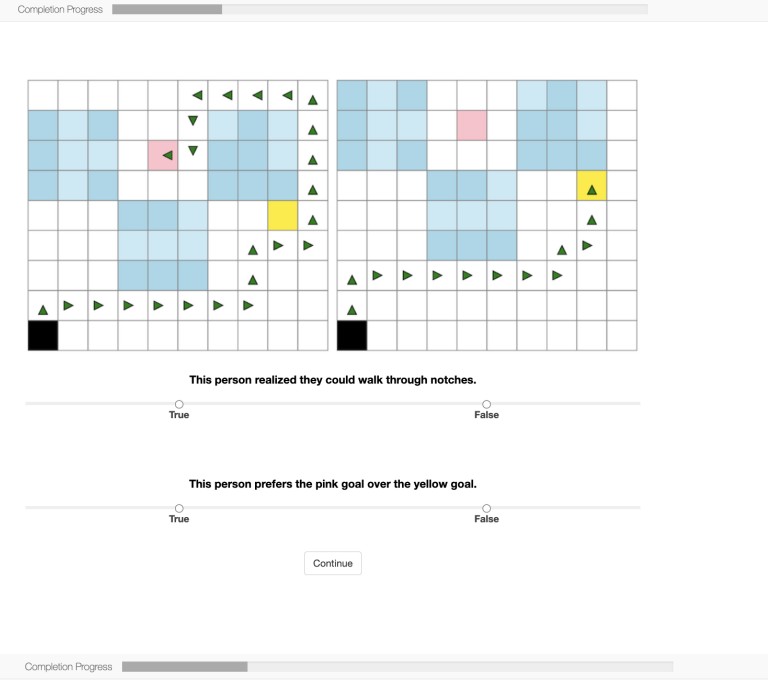

**This person realized they could walk through notches.**

○ True          ○ False

**This person prefers the pink goal over the yellow goal.**

○ True          ○ False

Continue

Now that you have had the chance to practice, let's begin with some real paths.

Next >

Completion Progress

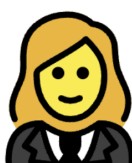

This is Janet. Here is Janet's path through the **1st** grid.

Next >

Completion Progress

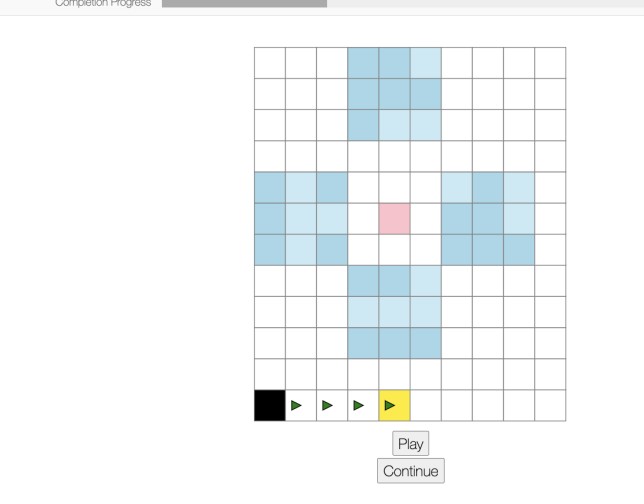

Play

Continue

Completion Progress

Now let's see Janet's path through the **2nd** grid.

Next >

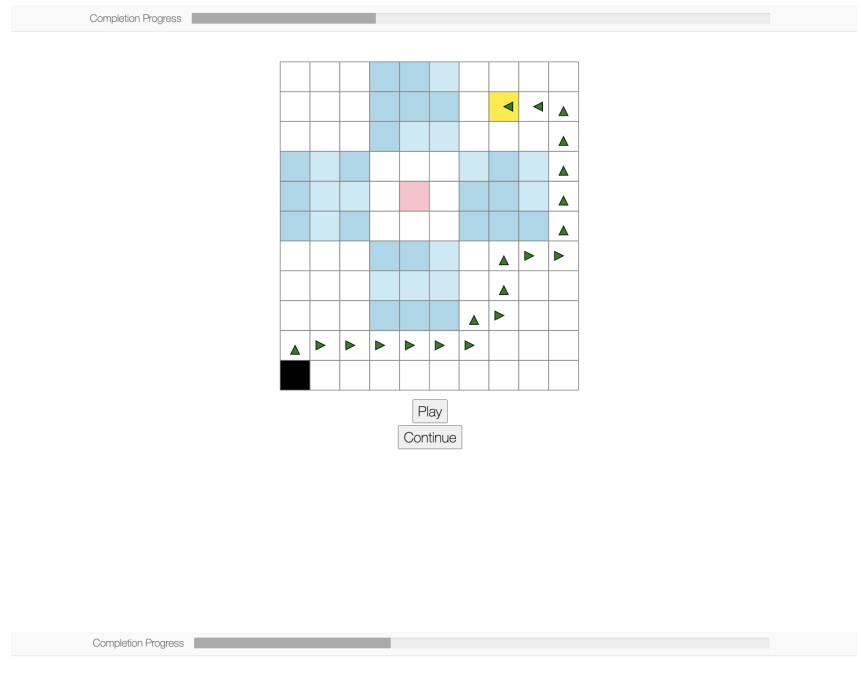

Completion Progress

Now let's see Janet's path through the **3rd** grid.

Next >

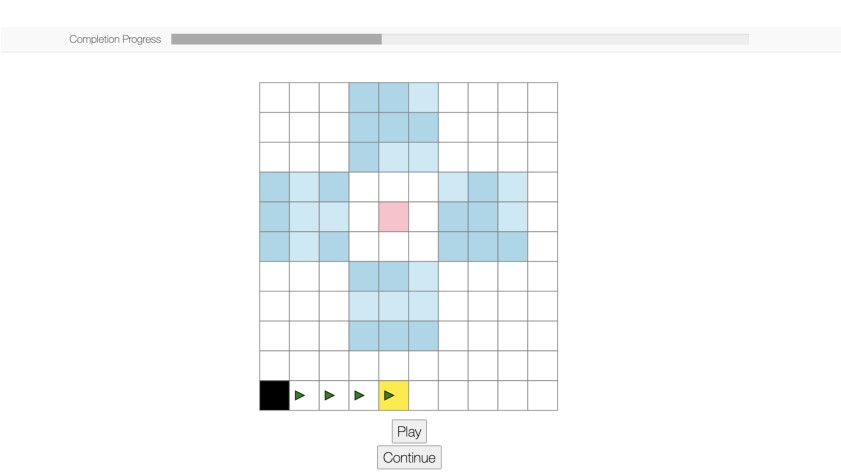

Completion Progress

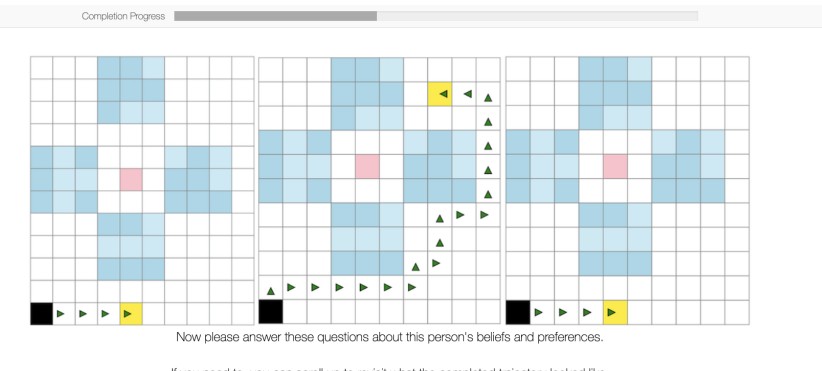

Now please answer these questions about this person's beliefs and preferences.

If you need to, you can scroll up to revisit what the completed trajectory looked like.

Next >

Completion Progress

**This person realized they could walk through notches.**

○ True          ○ False

**This person prefers the pink goal over the yellow goal.**

○ True          ○ False

Continue

Completion Progress

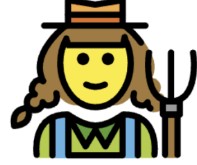

This is Andrea. Here is Andrea's path through the **1st** grid.

Next >

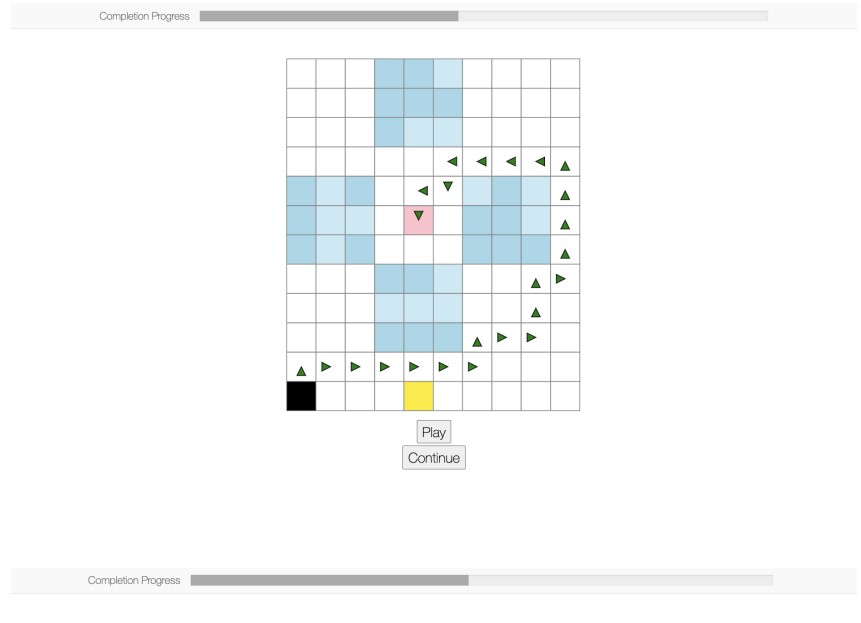

Now let's see Andrea's path through the **2nd** grid.

Next >

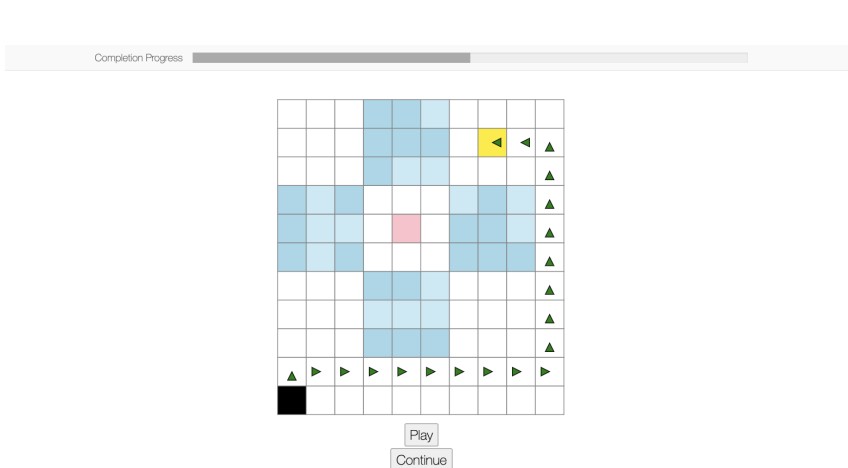

Completion Progress

Now let's see Andrea's path through the **3rd** grid.

Next >

Completion Progress

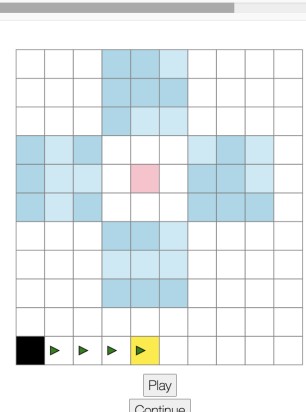

Play

Continue

Completion Progress

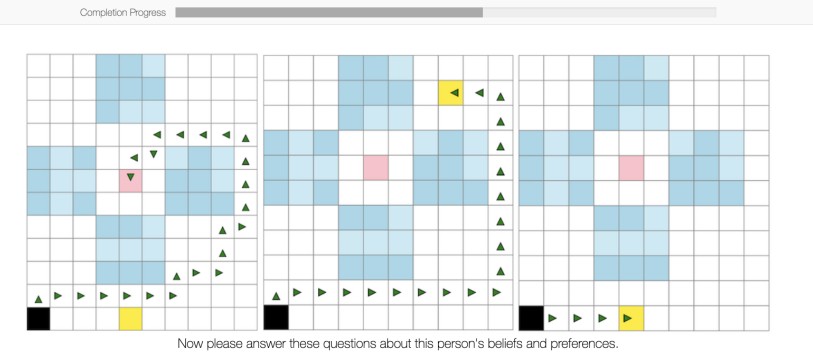

Now please answer these questions about this person's beliefs and preferences.

If you need to, you can scroll up to revisit what the completed trajectory looked like.

Next >

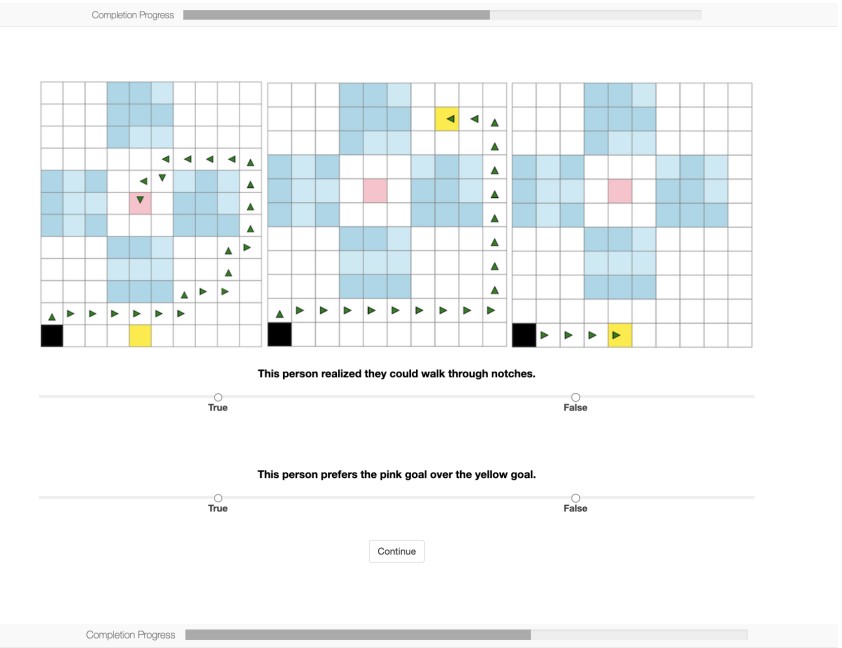

Completion Progress

**This person realized they could walk through notches.**

○ True          ○ False

**This person prefers the pink goal over the yellow goal.**

○ True          ○ False

Continue

Completion Progress

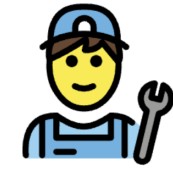

This is David. Here is David's path through the **1st** grid.

Next >

Completion Progress

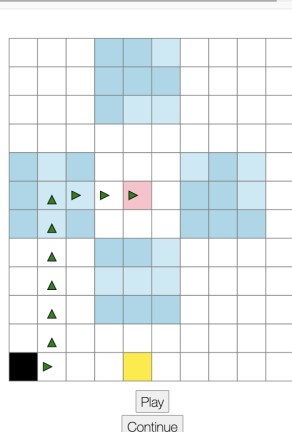

Play

Continue

Completion Progress

Now let's see David's path through the **2nd** grid.

Next >

Completion Progress

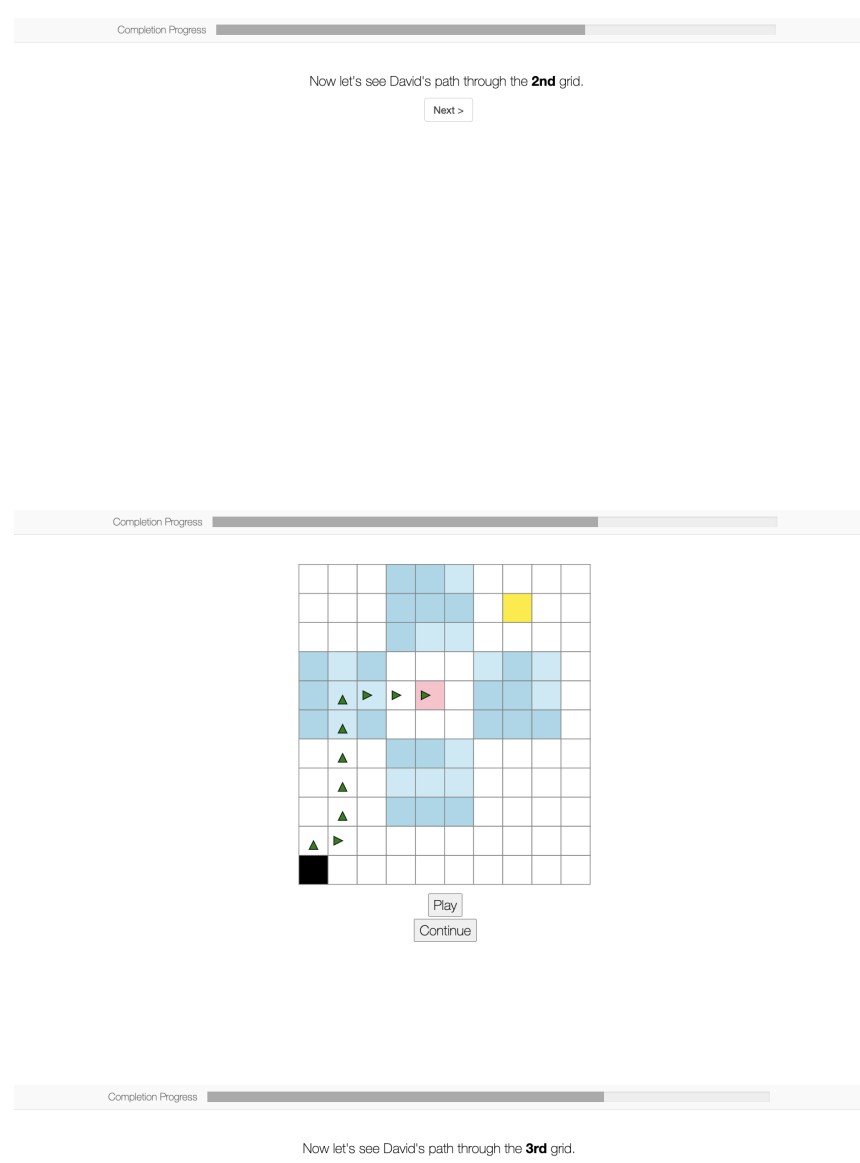

Play

Continue

Completion Progress

Now let's see David's path through the **3rd** grid.

Next >

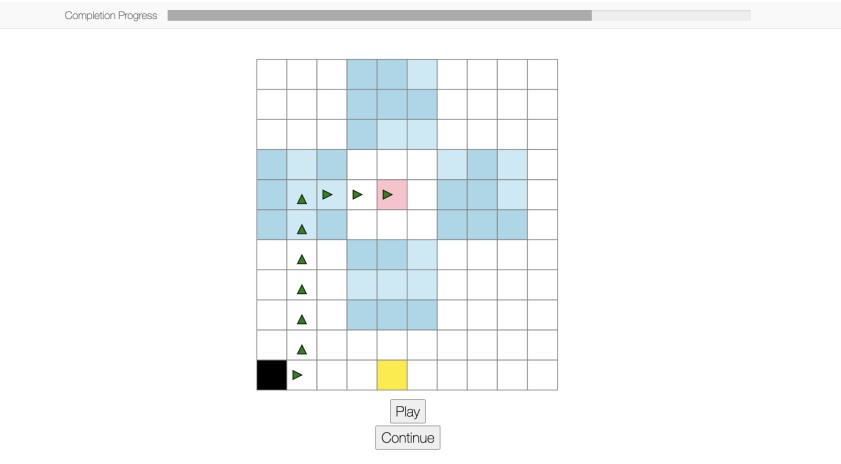

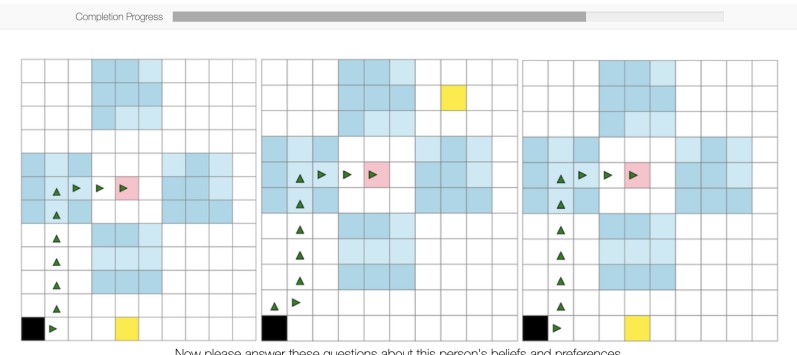

Now please answer these questions about this person's beliefs and preferences.

If you need to, you can scroll up to revisit what the completed trajectory looked like.

Next >

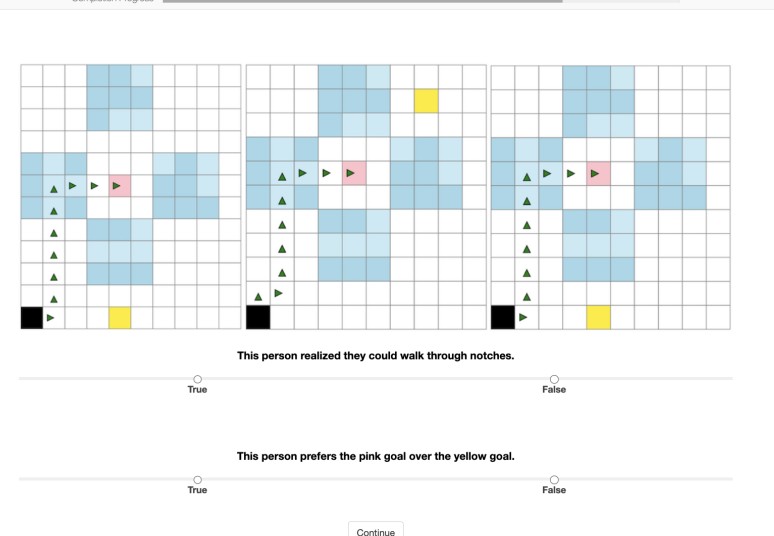

**This person realized they could walk through notches.**

○ True          ○ False

**This person prefers the pink goal over the yellow goal.**

○ True          ○ False

Continue

Completion Progress

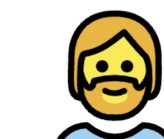

This is Jack. Here is Jack's path through the **1st** grid.

Next >

Completion Progress

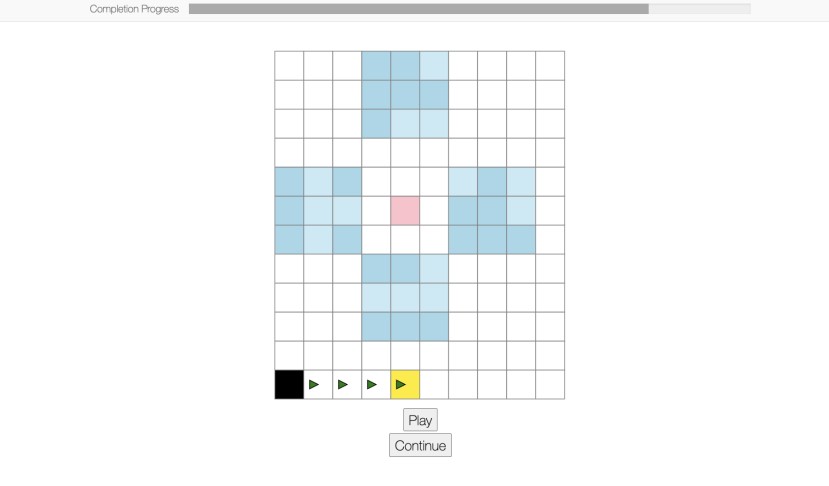

Play

Continue

Completion Progress

Now let's see Jack's path through the **2nd** grid.

Next >

Completion Progress

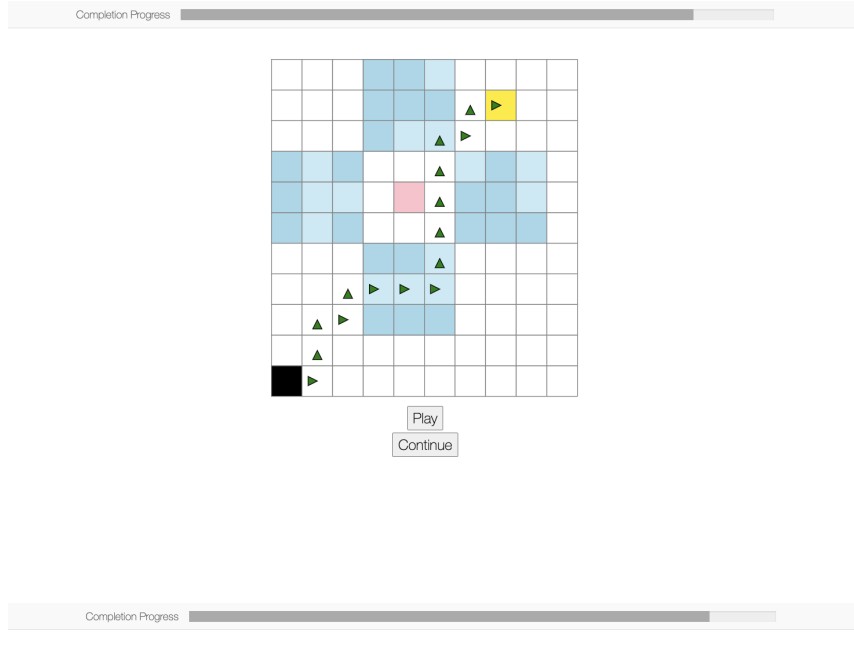

Play

Continue

Completion Progress

Now let's see Jack's path through the **3rd** grid.

Next >

Completion Progress

Play

Continue

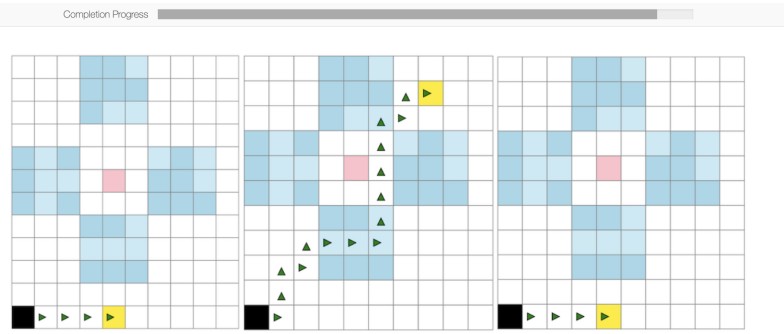

Now please answer these questions about this person's beliefs and preferences.

If you need to, you can scroll up to revisit what the completed trajectory looked like.

Next >

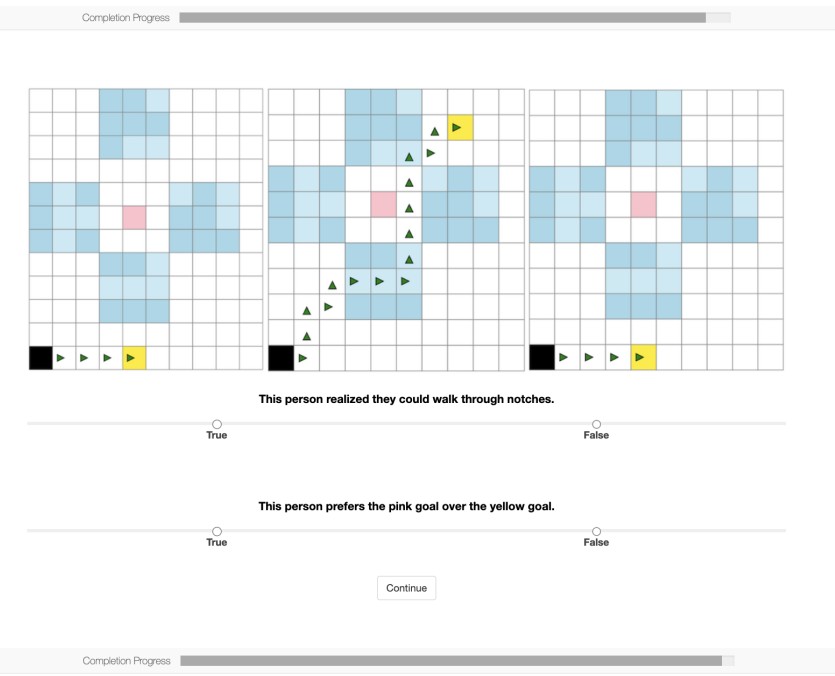

**This person realized they could walk through notches.**

○ True          ○ False

**This person prefers the pink goal over the yellow goal.**

○ True          ○ False

Continue

Thank you for participating!

Next >

