# OpenReview forum: "Concept Alignment as a Prerequisite for Value Alignment"
_ICLR.cc/2024/Conference — Submitted to ICLR 2024_

### Official Review · Reviewer_ouXJ · 2023-10-30

**Soundness:** 3 good
**Presentation:** 3 good
**Contribution:** 2 fair
**Rating:** 3
**Confidence:** 3

**Summary:**

In this paper, the authors analyze the value alignment problem from the conceptual alignment perspective. Inspired by recent theories on the cognitive understanding of users in problem-solving, a framework is proposed to solve the Inverse Reinforcement Learning (IRL) problem. The method builds on a bayesian formulation of the learning objective, which considers the conceptual understanding of different users in planning. The experiments are conducted over a simple planning maze, taken from previous works in cognitive science.

**Strengths:**

The authors address the failure of value alignment in IRL as an issue of concept misalignment. This is a new idea that could lead to potential benefits in learning personalized policies that are aligned with end users' concepts. The paper is well-written and clear and draws this important connection with previous work in cognitive sciences.

Overall, the idea, while simple, leads to sensible improvements compared to other strategies that are agnostic to users' concepts. The authors verified their method also via user studies which confirms the superiority of the proposed method. Appreciably, the IRL formulation taking into account the user construal attains a higher positive correlation with the human inference reward.

In conclusion, the authors motivate and prove that the integration of the users' construals has its own merits and deserves to be taken into account for future improvements in the setting of IRL.

**Weaknesses:**

The idea of connecting IRL with a construal model of the user is interesting but by far a milestone proposed in a series of previous papers on the subject. In the related work section, it is mentioned that Ho and Griffiths (2022) treated the problem in IRL. I understand that the fundamental contribution of the paper is essentially introducing (Eq. 3). The user concepts are already known beforehand and treating the hard case, where they are discovered or taken from a potentially big vocabulary, is only mentioned as future work for benchmarking this method. It is not clear what could be potentially cases solved by this approach, which requires more scrutiny.

Moreover, no details are provided on how Eq. 3 is implemented and the experiments are entirely synthetic, which leaves open how the framework could be adapted and what the impact would be in real-world cases where users' construal may change more. How do you estimate $P(R, \hat T)$? Is it already known beforehand? It is also not entirely clear what concepts $\hat T$ encode, are they semantical concepts related to some properties of the world (specific to one user) or are they just non-interpretable values reflecting the state of the user? Are the user concepts causal in nature, or belonging to some ontology? How is this related to the standard use of concepts in explainable AI, e.g. [1,2,3]?

My feeling is that this work goes in the right direction but lacks a challenging case study for the proposed method which renders the contribution limited for a paper in this venue. It is not mentioned whether the code will be also publicly available.

[1] P. W. Koh et al., Concept Bottleneck Models, ICML (2020) \
[2]  B. Kim et al. "Interpretability beyond feature attribution: Quantitative testing with concept activation vectors (tcav)." ICML  (2018) \
[3] S. Kambhampati et al., Symbols as a Lingua Franca for Bridging Human-AI Chasm for Explainable and Advisable AI Systems, AAAI (2022) (here referred to as symbols)

**Questions:**

I asked questions in the weaknesses section.

---

> ### Author Response · Authors · 2023-11-22
> **Author response to Reviewer ouXJ**
>
> We thank the reviewer for their feedback. It is quite true that our experiment is run on a simplified setting and not open-world settings. We use a simplified setting intentionally for two main reasons: First, we intended to demonstrate in an unambiguous and interpretable manner the limitations of value alignment as it is currently framed in the literature (often in simplified settings—see, for example, see Hadfield-Menell et al. 2016; Sumers et al., 2022), and to show how incorporating concept alignment could address those limitations. This is a necessary prerequisite for scaling up to more complex settings and a standard research strategy in AI safety and alignment research (see previous citations). Second, a major component of our contribution is the comparison of our model with human judgments that were experimentally elicited. By focusing on a simplified, controlled set of scenarios, we can make more direct comparisons between the formalism and behavioral data.
>
> Within this scope, we demonstrate the potential negative impact of concept-level misalignment if it is not taken into account, correct for it, and show human reasoning does take into account concept misalignment when inferring values. We know that it is necessary to both demonstrate the negative impact of concept misalignment and correct for it in real-world settings as well; in fact, this is the motivation for our work. Demonstrating this misalignment thoroughly in a simplified setting highlights the exact nature of the problem, and lays the groundwork for demonstrating it at scale and in the real world.
>
> Correcting for the problem in a measurable way, which is afforded to us by the closed nature of the simplified setting, allows us to start the critically important discussion of how it can be corrected in larger-scale, open-world settings. It will take time and experimentation to demonstrate the impact of concept misalignment on value misalignment for these large-scale, real-world settings, and even more time and discussion to correct for the misalignment in real-world settings. Each real-world setting comes with significant contextual background that must be taken into account when defining what “alignment” means in that setting, and how much alignment is needed. For example, scaling up concept alignment to a healthcare diagnostic algorithm would not require agent-based RL at all, and would not be implemented in an IRL setting. Rather it would be about the medical imaging artifacts that lead to diagnostic decisions, the words human physicians use to describe those artifacts, and how to align those concepts to the diagnostic algorithm which analyzes the same medical image and suggests a course of action, like surgery. This type of concept alignment would be a necessary prerequisite to discussing surgery based on what the patient values most (mobility, lack of pain, effect on other organs, long-term psychological impact of the procedure). We have significantly revised our “Societal Impacts” and “Limitations and Future Work” sections (6.2-6.3) to discuss this as suggested in the reviewer’s feedback.
>
> The code and data are indeed publicly available, linked here: https://osf.io/hd9vc/?view_only=967b0c2f981d4a87bf4d21ff818f1322
>
> To answer the reviewer’s questions regarding the translation of Eqn 3 to code:
> 1. $\tilde{T}$, the demonstrator’s construed task dynamics, encodes blocks and notches as shown in Figure 4. If the demonstrator’s construal is that they are paying attention to notches, their MDP reflects this based on which blue tiles seem traversable to them (notches) and which don’t (blocks). If they are not paying attention to notches, then their MDP reflects this with all blue squares as blocks.
> 2. The prior $P(R, \tilde{T})$ is uniform.
> We have added text to the updated paper clarifying both of these points, thank you for the questions.
>
> References
>
> Hadfield-Menell, D., Russell, S. J., Abbeel, P., & Dragan, A. (2016). Cooperative inverse reinforcement learning. Advances in neural information processing systems, 29.
>
> Sumers, T., Hawkins, R., Ho, M. K., Griffiths, T., & Hadfield-Menell, D. (2022). How to talk so AI will learn: Instructions, descriptions, and autonomy. Advances in Neural Information Processing Systems, 35, 34762-34775.

---

### Official Review · Reviewer_LqqC · 2023-11-01

**Soundness:** 2 fair
**Presentation:** 2 fair
**Contribution:** 1 poor
**Rating:** 1
**Confidence:** 4

**Summary:**

The work motivates the importance of modeling human mental model of the task (or the dynamics function they would use) to generate demonstrations for an agent while the agent attempts to perform IRL. They present this as "Inverse Construal" problem and provide a grid world based example for the same. They argue, citing prior works, that a way human dynamics function may be different from the agent dynamics function is because humans may simplify the dynamics so that they are able to plan / generate demonstrations easily. They conduct a subject study to highlight their arguments for this domain.

**Strengths:**

The example is helpful in understanding the arguments made in the paper.

The paper presents a useful subject-study that establishes the need for modelling human dynamics along with learning reward models in the context of IRL.

**Weaknesses:**

It seems that the work is pushing for “IRL agents attempting to learn from human trajectories should take into account human mental model of the task”. There is a considerable body of work that highlights the importance of modelling human mental models for behavior synthesis (as an example [1]) in automated planning. That is, taking into account human mental models (such as the transition function being used by them) for agent tasks like behavior synthesis, goal recognition, intention prediction etc. is already well motivated (and the current discussion seems to rediscover this for IRL).

Even in the context of IRL, the notion of the correspondence problem (see Related work in [4]) is very related here (not discussed in the paper). The key idea is the dynamics of the demonstrations are different than the dynamics of the agent and the mis-match causes issues with leveraging the demonstrations. In this work the mis-match is motivated through a specific situation where the human demonstrator plans on “simplified dynamics“, however the formulation does not make this distinction. With respect to the formulation the authors assume that the demonstrator has a different dynamics $\tilde{T}$ where there is no formal restriction on  $\tilde{T}$ and a loose specification that  $\tilde{T}$ is ”simpler or easier to solve“. Typically correspondence problem has been viewed through the lens that the demonstrations were collected in a different domain and the agent is acting in a different domain (for example demonstrations is a real human providing robot arm movement and the agent is working in a simulated environment), which still satisfies the problem formulation in this paper, that the dynamics are different.

There are missing arguments on what  $\tilde{T}$ can be. For example, it seems that it has to be defined over the same states and actions as the agent dynamics function  ${T}$ (through result in section 3.4 and equation 4). This implies that human mental model (or the dynamics they are using to come up with a plan) cannot have arbitrarily different state representations (which I believe comes from the authors description on “simpler and easier to solve”, but the formalism is unclear from text).

The example considered by the authors, as I understand, assumes that the demonstrator has a different dynamics function as a consequence of state-aliasing [2, 3] or perceptual aliasing (i.e. they mix up the light blue and blue cells etc.). These concepts are well studied in sequential decision making but the current manuscript fails to make essential connections highlighting limited literature review.

Authors have not reported IRB and subject study details like demographics. I am flagging the work for ethics review.

[1] Chakraborti, T., Kulkarni, A., Sreedharan, S., Smith, D. E., & Kambhampati, S. (2019). Explicability? legibility? predictability? transparency? privacy? security? the emerging landscape of interpretable agent behavior. In Proceedings of the international conference on automated planning and scheduling (Vol. 29, pp. 86-96).

[2] Gopalakrishnan, S., Verma, M., & Kambhampati, S. (2021, June). Synthesizing policies that account for human execution errors caused by state aliasing in markov decision processes. In ICAPS 2021 Workshop on Explainable AI Planning.

[3] McCallum, A. K. (1996). Reinforcement learning with selective perception and hidden state. University of Rochester.

[4] Cao, Z., Hao, Y., Li, M., & Sadigh, D. (2021). Learning feasibility to imitate demonstrators with different dynamics. arXiv preprint arXiv:2110.15142.

**Questions:**

Please refer to "Weaknesses" section.

**Details Of Ethics Concerns:**

I would request the authors to provide details on whether the study was approved by an IRB and details on subject study such as participant demographics, criteria for hiring subjects etc.

---

> ### Author Response · Authors · 2023-11-22
> **Author response to Reviewer LqqC**
>
> We thank the reviewer for their feedback. It is quite true that this work is not intended to propose an improved IRL algorithm or framework. Rather the purpose of the work, as an AI safety contribution, is to use an existing IRL framework to experimentally demonstrate the consequences of concept-level misalignment to value-level misalignment between humans and machines, and also demonstrate how this misalignment can be remedied if concepts are taken into account.
>
> To this end, our goal is to provide an empirical starting point for a discussion on how we can incorporate concept-level alignment across forms of AI. Many instances where concept alignment is required will not involve RL at all. For example, scaling up concept alignment to a healthcare diagnostic algorithm would not require agent-based RL at all, and would not be implemented in an IRL setting. Rather it would be about the medical imaging artifacts that lead to diagnostic decisions, the words human physicians use to describe those artifacts, and how to align those concepts to the diagnostic algorithm which analyzes the same medical image and suggests a course of action, like surgery. This type of concept alignment would be a necessary prerequisite to discussing surgery based on what the patient values most (mobility, lack of pain, effect on other organs, long-term psychological impact of the procedure). Due to the breadth of AI safety concerns and applications, it is not possible to exhaustively address concept alignment in a single paper; rather, it will require a larger discussion around how to achieve concept-level alignment between humans and AI.
>
> We do of course have IRB approval, and a walkthrough of our study is included as an appendix to the original submission. The study was conducted through the commonly-used research participant recruitment platform Prolific, and participants were from the US and UK. Participants were not filtered by other demographics or through any other questions. We are happy to share deanonymized IRB documentation with ethics reviewers if appropriate.
>
> Finally, we thank the reviewer for drawing our attention to this important body of work in automated planning on modeling human mental models. In the updated version of the paper, we now reference this literature. Our work builds on these important contributions in two ways that have not been previously approached in the literature (as far as we are aware). First, we benchmark discrepancies in mental models between standard IRL and real humans using controlled behavioral studies that builds on an established experimental paradigm from the psychology literature. Second, we propose a general inferential framework for approaching this problem grounded in recent work on computational cognitive science. As far as we are aware, this has not been previously explored in the literature. For example, although the workshop paper that the reviewer references by Gopalakrishnan et al (2021) does report comparisons with humans, it makes only coarse-grained predictions about policy imitation (e.g., in complex versus simple policies), whereas our account explicitly considers the role of mental models via inference about the process of construal. Our work takes a computational cognitive science perspective to ensure that discussion of value alignment in AI safety remains concretely grounded in human behavior.

---

> ### Comment · Reviewer_LqqC · 2023-11-22
> **Response to authors**
>
> I appreciate author's response to my concerns. [I am not able to see the revised pdf, as open review still shows that there are no revisions.]
>
> After the author's response, I feel my understanding of the work was correct and I remain on my point on limited novelty as the need for mental modeling has been established in several prior works. I would appreciate if the authors could address my concerns on their assumptions on the transition dynamics function [see para 2 and 3 of my review].
>
> I am happy to read that the authors followed an approved IRB protocol. They should follow HRI works to report human subject study information [for double-blind submissions] and provide details such as demographics, participant hiring information etc. for reproducibility. This becomes all the more important as the authors feel their primary contribution is to "benchmark discrepancies in metal models .... using controlled behavioral studies".
>
> I will maintain my score based on our existing discussion and after reading comments by other reviewers. I feel the work can substantially benefit from making it explicit what their claims are and clearly contrast with prior work. The current study is definitely a first step towards establishing the need for mental modeling and the authors can explore its utility further in their setting of IRL or AI safety.

---

> > ### Author Response · Authors · 2023-11-23
> > **Author response to comment by Reviewer LqqC**
> >
> > We thank the reviewer for their response and their time. The revised pdf was uploaded before the author’s response was posted, and it has been accessible throughout this time on OpenReview. We request the reviewer to please let us know if they are still unable to access the revised pdf, as this would indicate there is a technical issue beyond our control. To further address the reviewer’s question about dynamics: $\tilde{T}$, the demonstrator’s construed task dynamics, encodes blocks and notches as shown in Figure 4. If the demonstrator’s construal is that they are paying attention to notches, their MDP reflects this based on which blue tiles seem traversable to them (notches) and which don’t (blocks). If they are not paying attention to notches, then their MDP reflects this with all blue squares as blocks. This abstraction represents the idea that “people are resource-rational---that is, they think and act rationally, but are subject to cognitive limitations on time, memory, or attention” (Section 2, Related Work). We hope this clarifies any lingering questions about the dynamics, and we have already added clarifying language about $\tilde{T}$ to the updated paper uploaded yesterday.
> >
> > While we are disappointed that the reviewer maintains a score of 1 for this work, we would be grateful if the reviewer could elaborate which previous works they feel have already addressed the issues in this paper. We are not aware of any other works which both highlight the challenges of concept misalignment and simultaneously show how direct efforts at concept alignment can be used to remedy value misalignment. The authors would very much benefit from learning about previous works which do both of these things.

---

> > > ### Comment · Reviewer_LqqC · 2023-11-23
> > > **Response to authors**
> > >
> > > It seems I am still unable to see the updated pdf. I am checking with "Revisions" on the submission which says that there are no revisions. In any case, I assure the authors that I will check again later / with other reviewers as needed to make sure I do not miss any details critical to my review.
> > >
> > > Meanwhile my question on transition dynamics is as given in the original description. It is possible that authors have improved the formalism to answer that, but it is not present in their rebuttal text. From the author's rebuttal it seems like a their description of the transition dynamics as what the demonstrator is paying attention to, is again, similar to the arguments made in the several works arguing for mental modeling of other agents.
> > >
> > > While the rebuttal period is about to end, I would suggest the authors to view relevant works in Kulkarni et al, and Dragan et al on Plan Legibility which by definition takes into account the human mental models. Further, they should also view several works on the correspondence problem. The current work motivates the need to model the transition dynamics of the human (which the authors rediscover as construals for their IRL setting). I feel there are fundamental similarities between their work and those in AI Planning etc. that should be highlighted and can make up for a stronger contribution that clearly highlights their contribution.

---

> > > > ### Author Response · Authors · 2023-11-23
> > > > **Author response to comment by Reviewer LqqC**
> > > >
> > > > We thank the reviewer for their response. We were not able to find the texts the reviewer mentioned (“Kulkarni et al, and Dragan et al”), nor do they seem to be referenced in any previous discussion with this reviewer. Perhaps the reviewer is referring to Chakraborti et al. which was referenced in their original review? We did, however, make an attempt to understand the reviewer’s perceived link between plan legibility and our work in alignment; to the best of our understanding, “Legible planning is the creation of plans that best disambiguate their goals from a set of other candidates from an observer’s perspective.” [1] The previous work in creating legible plans thus seems directly related to larger goals in agent interpretability and human-robot interaction. We share these larger goals. However, our work does not involve trying to make an AI system choose plans that are more interpretable to a human observer, but rather to make a human’s behavior more interpretable to an AI system by imbuing the AI system with a way of modeling concepts the human may be using to understand the world. Importantly, our work argues explicitly for a shared understanding of *concepts* between humans and machines, which, to our knowledge, plan legibility work does not.
> > > >
> > > >
> > > > Similarly, we made an attempt to understand the reviewer’s suggestion to delineate our work from other work regarding the correspondence problem. The reviewer says, *“there is no formal restriction on $\tilde{T}$ and a loose specification that $\tilde{T}$ is ‘simpler or easier to solve’. Typically correspondence problem has been viewed through the lens that the demonstrations were collected in a different domain and the agent is acting in a different domain (for example demonstrations is a real human providing robot arm movement and the agent is working in a simulated environment).”* We clarified the restrictions on $\tilde{T}$ in our previous response regarding dynamics, and elaborate here. First, $\tilde{T}$ explicitly encodes the concepts of blocks and notches as we described in our previous response, and for the purposes of our experiments it is limited to encoding only those features. It is neither unconstrained nor is it loose. Our revised paper and past responses clarify precisely what $\tilde{T}$ encodes. Secondly, our work does not involve demonstrations collected in one domain and the agent acting in another, per the correspondence problem. Rather, it involves an AI agent trying to understand different humans acting in the *same* domain, but behaving differently based on their different conceptual understandings of that domain, as dictated by “cognitive limitations on time, memory, or attention” that all humans are subject to (Section 2, Related Work).
> > > >
> > > > Perhaps another way to frame this in the reviewer’s own terms is through language from one of the papers the reviewer referenced in their original review, Chakraborti et al., which defines “behaviors” as *“one particular observed instantiation of a plan or policy. In particular, a plan – which can be seen as a set of constraints on behavior – engenders a candidate set of behaviors [Kambhampati et al., 1996] some of which may have certain interpretable properties while others may not.”* Our work argues that human-AI alignment must be at the *conceptual* level, and the human and the machine would benefit from first having a shared understanding of the concepts used in the decision-making process.
> > > >
> > > > In other words, the shared *concepts* are the “interpretable properties” we argue for in plans *and* behaviors produced by AI systems. To this end, our results show one concrete way in which shared concepts can improve AI alignment and safety. This is something which has not been shown or discussed before, to our knowledge, in any of these previous works.
> > > >
> > > > We do hope the reviewer is able to find our revised paper in order to make an accurate assessment of the work. We are happy to provide any assistance we can in this process.
> > > >
> > > > References
> > > >
> > > > [1] Persiani, M., Hellstrom, T. “Probabilistic Plan Legibility with Off-the-shelf Planners”

---

### Official Review · Reviewer_thM9 · 2023-11-02

**Soundness:** 3 good
**Presentation:** 3 good
**Contribution:** 3 good
**Rating:** 3
**Confidence:** 5

**Summary:**

This paper highlights the importance of considering a human's mental approximations (described as "construals") when making inferences about human preferences from their observed behavior.  As humans often rely on approximate models of the real world when planning their actions, evaluating potential preference models under the assumption that they plan under an exact world model may lead to incorrect inferences about their true preferences.  This in turn may lead to misalignment between an AI's future actions and what the human would have expected.

They formalize this problem in terms of Bayesian inverse reinforcement learning, where the "true" dynamics of the environment (which the AI knows exactly) are replaced with multiple possible approximations, which are jointly inferred with the reward function encoding the human's preferences.  Their main contribution is a set of experiments demonstrating that their "construal" IRL model matches the inferences of human subjects far better than standard Bayesian IRL in a navigation task where an approximate model leads to behavior that is very different from optimal behavior under the true model.

**Strengths:**

The primary strength of the paper is in highlighting the importance of concept alignment in efforts to achieve human-AI alignment.  This issue takes on far greater significance today than it did a few years ago, as AI agents trained with human feedback and demonstrations are now widely deployed in the real world.  The work demonstrates that even in simple settings, failure to account for mental approximations can lead to catastrophic misalignment.  The key takeaway is that we must be careful when learning from humans that we account for limitations in the human's knowledge of the world, and how this might affect their behavior relative to their preferences.

The paper also draws important links between existing psychological research on human conceptualization and planning, and the problem of human-AI value alignment.

**Weaknesses:**

My main concern is that there are conceptual barriers to applying the insights of this work to algorithms that scale to real-world problems.  Bayesian IRL can be viewed as a "regularized" form of behavioral cloning, where the preference for policies that are optimal under high-probability reward functions improves generalization from limited amounts of data.  The inference model proposed here retains this advantage because the space of reward functions and concepts is tightly constrained.  Scaled-up, however, both the reward model and the approximate planning model would need to be far more complex, to the point where we would not expect (an approximation of) Bayesian IRL to be any more sample efficient that behavioral cloning with a similarly complex policy model.  Put another way, given enough data and a sufficiently flexible reward model, we would expect "exact" Bayesian IRL to be able to predict human behavior as well as "construal" BIRL, with the difference in sample complexity becoming less significant as we scale up to more complex tasks.

While I wouldn't expect the paper to solve these issues itself, a deeper discussion of these potential limitations would be useful to the reader.  It would also have been nice to see connections drawn between this work and more scalable approaches to learning from demonstration (e.g., Ho and Ermon, 2016)

The other weakness with the work is that the theoretical model is not itself particularly novel.  Essentially they do Bayesian IRL where the parameters of both the reward function *and* the dynamics model are inferred from human behavior.  A number of previous works have used essentially the same model (e.g., Herman et al. 2016)

The authors should reference previous work in this space, and highlight how the motivations of this work differ from those of previous works with similar mathematical models.

References:
1. Ho, Jonathan, and Stefano Ermon. "Generative adversarial imitation learning." Advances in neural information processing systems 29 (2016).
2. Herman, Michael, et al. "Inverse reinforcement learning with simultaneous estimation of rewards and dynamics." Artificial intelligence and statistics. PMLR, 2016.

**Questions:**

1. Did any of the human subjects experiments evaluate human inference with a subset of the trajectories?  It seems possible that humans might make the same inferences about preferences and concepts without sufficient examples to rule out alternative hypotheses.  For example, differences between experiments might que them to provide different answers than they did in previous experiments.
2. A minor point, but were any participants rejected because they failed to correctly understand the "notch" concept themselves?

---

> ### Author Response · Authors · 2023-11-22
> **Author response to Reviewer thM9**
>
> We thank the reviewer for their feedback. It is true that this is not intended to scale in its current form, but rather demonstrate concretely the limitations of value alignment as it is currently discussed in the literature, and to show how incorporating concept alignment could address those limitations. To understand the limitations of value alignment at scale, the AI safety community will need to study it across many different contexts and disciplines, of which this paper addresses only one.
>
> What these limitations look like vary greatly depending on the context of the AI being used, and the degree of human-AI alignment needed in each of those contexts. For example, scaling up concept alignment to a healthcare diagnostic algorithm would not require agent-based RL at all, and would not be implemented in an IRL setting. Rather it would be about the medical imaging artifacts that lead to diagnostic decisions, the words human physicians use to describe those artifacts, and how to align those concepts to the diagnostic algorithm which analyzes the same medical image and suggests a course of action, like surgery. This type of concept alignment would be a necessary prerequisite to discussing surgery based on what the patient values most (mobility, lack of pain, effect on other organs, long-term psychological impact of the procedure). If the AI is not able to honestly and concretely align concepts with the patient or the doctor, and meaningfully explain why a diagnostic judgment was made, the resulting medical decision is likely to be value misaligned with the patient in critical ways.
>
> In contrast, scaling up concept alignment to something like deep RL could be much more similar to our Bayesian IRL example, with the values explicitly tied to the reward functions. We anticipate and encourage this breadth of work studying concept alignment and the resulting effect on value alignment, or misalignment, as a critical step forward in AI safety discussions.
>
> We have significantly revised our “Societal Implications of Concept Misalignment” section and “Limitations and Future Work” section (6.2-6.3) to discuss these implications, per the reviewer’s suggestion. We have also highlighted how our motivations differ from the previous works with similar mathematical models mentioned in the review, and the updated paper now references these works, thank you for bringing them to our notice.
>
> Answers to questions:
> 1. All human subjects were shown the full set of trajectories, which were designed to have sufficient examples to rule out alternative hypotheses. The human subjects and the IRL models were of course shown the same set of trajectories.
>
> 2. Participants were not rejected/excluded for failing to correctly understand the "notch" concept.

---

> > ### Comment · Reviewer_thM9 · 2023-11-22
> > **Response to authors**
> >
> > Thank you for taking the time to respond to my comments and questions.
> >
> > I tend to agree with the other reviewers that most of the ideas presented in this paper have appeared elsewhere.  A number of previous works have considered the challenge of imitation learning in settings where the human's understanding of the environment (either the dynamics or the current state) may not match that of the AI.  The conceptual contribution of the paper is therefore very limited.
> >
> > The main contribution of the work is really the experimental results showing that humans tend to make "construal" inferences from observations of other's behaviour, and are able to recognise mental approximations used by others.  These results are also somewhat limited, as they only consider a single experimental task, and it remains possible that there are other explanations for the observed results.  Clearly demonstrating that humans reliably infer other's mental approximations would require a larger set of experiments, with an emphasis on identifying situations in which humans *fail* to make such inferences (for example, if the space of possible simplifications is not made explicit in the instructions for the experiment).
> >
> > Furthermore, I feel that there is something of a disconnect between the focus of discussion in the paper on the importance of accurate construal inference for human-AI alignment, and the experimental results which essentially go in the opposite direction (showing that humans themselves make construal inferences).  The real take-away from the paper is that AI agents need to consider how construal inferences might affect human's predictions of the AI's behaviour.  This could be an issue in human-robot interaction settings, for example.

---

> > > ### Author Response · Authors · 2023-11-23
> > > **Author response to comment by Reviewer thM9**
> > >
> > > We thank the reviewer for their response and their time. While we are disappointed that the reviewer has changed their previously positive score, we would be grateful if the reviewer could elaborate which previous works they feel have already addressed the issues in this paper. We are not aware of any other works which both highlight the challenges of concept misalignment in instances where “the human's understanding of the environment (either the dynamics or the current state) may not match that of the AI” and simultaneously show how direct efforts at concept alignment can be used to remedy value misalignment. The authors would very much benefit from learning about previous works which do both of these things.

---

### Official Review · Reviewer_cFhp · 2023-11-03

**Soundness:** 3 good
**Presentation:** 4 excellent
**Contribution:** 3 good
**Rating:** 6
**Confidence:** 2

**Summary:**

The authors look at the benefits of implementing concept simplification strategies into inverse reinforcement learning (this yields "inverse construal").  The key message is that if an IRL algorithms that fail to model what the demonstrator knows risk failing to understand their reward function.  A bound is provided for the gap in performance between a construal-aware estimate and an entropy-regularized one.  The authors demonstrate this issue in a novel synthetic setting and with user-collected data.

**Strengths:**

**Originality**: To the best of my knowledge, the idea of modeling construals (what the demonstrator knows or is constrained by) is new in IRL.

**Clarity**: The text is crystal clear and very easy to follow.  The examples are well constructed.  All critical steps are properly formalized, and the notation is consistent.

**Quality**: The related work section is very well done.  The experimental setup also seems reasonable -- but I am not an expert, so I wouldn't be able to tell whether there are implicit biases in the data collection.

**Significance**: I think the high-level message is very much important and I agree with it.  I think the message is well worth discussing at the conference.

**Weaknesses:**

**Clarity**: I am confused about the usage of the notion of "concept" in this context.  In explainable AI, concepts refer to high-level representations (presumably interpretable) of a given input to be explained or otherwise processed.  In cognitive science and logic it has a similar meaning.  Here it is used as a synonym for knowlede or construal.  I would appreciate if the authors could clarify this in the introduction.

**Quality**: [Q1] The single biggest issue with the paper is that it considers only a rather toy synthetic setting.  I am not sure if this is common in the IRL literature.

**Questions:**

I would appreciate some clarification regarding Q1 above.

---

> ### Author Response · Authors · 2023-11-22
> **Author response to Reviewer cFhp**
>
> We thank the reviewer for their feedback. The notion of a “concept” is indeed broad, and construals here are meant to be one potential instantiation of what a concept could look like in a concrete, contained example. Per the reviewer’s suggestion, we have added a clarifying sentence to the introduction explaining that different construals are used to encode different conceptual understandings of the world.
>
> The reviewer is correct that we use a synthetic, simplified setting to study the problem of concept alignment. This was done intentionally for two main reasons. First, we intended to demonstrate in an unambiguous and interpretable manner the limitations of value alignment as it is currently framed in the literature (often in simplified settings—see, for example, see Hadfield-Menell et al. 2016; Sumers et al., 2022), and to show how incorporating concept alignment could address those limitations. This is a necessary prerequisite for scaling up to more complex settings and a standard research strategy in alignment research (see previous citations). Second, a major component of our contribution is the comparison of our model with human judgments that were experimentally elicited. By focusing on a simplified, controlled set of scenarios, we can make more direct comparisons between the formalism and behavioral data.
>
> To understand the limits of value alignment at scale, the AI safety community must study it across many different contexts and disciplines, of which this paper addresses only one. “Values” of AI systems are tricky to define, and an IRL setting gives us one very concrete way to define them (in terms of reward functions). We agree that it is very important to study AI “values” outside of such a controlled setting, and even more important that we also study concepts in tandem with values (and we hope this paper demonstrates why studying both together is important). We hope that our empirical framework provides a starting point for designing such studies.
>
> References
>
> Hadfield-Menell, D., Russell, S. J., Abbeel, P., & Dragan, A. (2016). Cooperative inverse reinforcement learning. Advances in neural information processing systems, 29.
>
> Sumers, T., Hawkins, R., Ho, M. K., Griffiths, T., & Hadfield-Menell, D. (2022). How to talk so AI will learn: Instructions, descriptions, and autonomy. Advances in Neural Information Processing Systems, 35, 34762-34775.

---

> > ### Comment · Reviewer_cFhp · 2023-11-23
> > **Reply to Authors**
> >
> > Thank you for the detailed response.  I agree that there is value in working with simplified settings, but it is also clear that - by constructions - these settings do not match the full complexity of real-world applications.  In this sense, I still think that working with *only* synthetic settings is a weakness.
> >
> > Regardless, considering how negative the other reviewers were concerning novelty, I will have to discuss with them before deciding whether and how to revise my score.

---

> > > ### Author Response · Authors · 2023-11-23
> > > **Author response to comment by Reviewer cFhp**
> > >
> > > We agree that working in a synthetic setting is a limitation. We believe this was necessary to meaningfully engage with the existing value alignment literature, and to concretely demonstrate the limitations and dangerous consequences of value alignment as it is currently discussed in the literature. We do not see how it is possible to scale such a study immediately to non-simplified real-world settings without substantial, community-wide further work across AI Safety (as described in our response above and in our revised paper), but we are open to and would encourage reviewers’ suggestions on how to do so if reviewers believe it is indeed immediately feasible. We see this instead as a longer-term goal, which the authors believe is very important and will continue working towards. As discussion closes, we would like to thank the reviewer once again for their thoughtful and constructive feedback throughout the review process, and for taking the time to understand the motivations and merits of the paper as an AI Safety contribution.

---

### Meta-Review · Area_Chair_aXWZ · 2023-12-05

**Metareview:**

Despite the paper's originality and clarity, and its valuable attempt to address value alignment in Inverse Reinforcement Learning by considering user concepts or construals, it falls short in several critical aspects, leading to a decision of rejection. While the idea of integrating user construals into IRL models is novel and the paper is well-written, it primarily considers a synthetic setting which limits its applicability to real-world scenarios. The work's incremental nature, with its reliance on existing techniques and concepts, further weakens its contribution. The lack of a challenging real-world case study and the absence of details on implementation and adaptation to dynamic real-world situations, where user concepts may change, are significant limitations. Overall, these shortcomings in scope, novelty, and ethical compliance outweigh the paper's strengths, leading to its rejection.

**Justification For Why Not Higher Score:**

The paper is rejected for the following reasons:

Limited Real-World Applicability: The focus on a synthetic setting greatly limits the paper's relevance and applicability to real-world IRL problems.

Incremental Nature: The work is seen as an incremental development over existing models, without offering substantial advancements or novel insights.

Lack of Real-World Case Study: There is no challenging real-world case study to demonstrate the practical effectiveness of the proposed method.

**Justification For Why Not Lower Score:**

N/A

---

### Decision · Program_Chairs · 2024-01-16

Reject